# SWE-smith:
# Scaling Data for Software Engineering Agents

**John Yang[1], Kilian Lieret[2], Carlos E. Jimenez[2], Alexander Wettig[2], Kabir Khandpur[3], Yanzhe Zhang[1], Binyuan Hui[4], Ofir Press[2], Ludwig Schmidt[1], Diyi Yang[1]**

[1]Stanford University    [2]Princeton University    [3]Indepedent    [4]Alibaba Qwen

## Abstract

Despite recent progress in Language Models (LMs) for software engineering, collecting training data remains a significant pain point. Existing datasets are small, with at most 1,000s of training instances from 11 or fewer GitHub repositories. The procedures to curate such datasets are often complex, necessitating hundreds of hours of human labor; companion execution environments also take up several terabytes of storage, severely limiting their scalability and usability. To address this pain point, we introduce SWE-smith, a novel pipeline for generating software engineering training data at scale. Given any Python codebase, SWE-smith constructs a corresponding execution environment, then automatically synthesizes 100s to 1,000s of task instances that break existing test(s) in the codebase. Using SWE-smith, we create a dataset of 50k instances sourced from 128 GitHub repositories, an order of magnitude larger than all previous works. We train `SWE-agent-LM-32B`, achieving 40.2% Pass@1 resolve rate on the SWE-bench Verified benchmark, state of the art among open source models. We open source SWE-smith (collection procedure, task instances, trajectories, models) to lower the barrier of entry for research in LM systems for automated software engineering. All assets are available at `https://swesmith.com`.

## 1   Introduction

Language Model (LM) agents, such as SWE-agent [49] or OpenHands [38], have made remarkable progress towards automating software engineering (SE) tasks, as tracked by benchmarks such as SWE-bench [18]. However, the most effective agents rely heavily on proprietary LMs. On the other hand, building open source LMs for SE remains bottlenecked by the lack of large-scale, high-quality training data. To keep open research relevant, it is critical to develop infrastructure for collecting software engineering training data at scale.

The current open-source software ecosystem offers two kinds of data sources to train LMs on SE tasks. One simple approach is to crawl pull requests (PRs) and issues from GitHub repositories. However, without execution environments or tests, these instances offer no reliable way of validating generated solutions, and LMs are limited to learning from the surface form of code [44] or via rewards based on superficial string similarity [41].

In contrast, SWE-bench provides reliable validation by running unit tests against proposed solutions. Another line of work has simply extended the SWE-bench collection strategy to a new set of repositories for training purposes [31]. This produces flexible environments for training and distilling LM agents, since we can generate agent trajectories and filter them based on the unit test results. However, the scalability of this approach is severely limited by the challenges associated with SWE-bench's collection strategy. SWE-bench's filtering process leaves only a small number of PRs that not only resolve a Github issue, but also make meaningful changes to unit tests. Also, setting up execution environments for each instance requires a substantial amount of human intervention.

39th Conference on Neural Information Processing Systems (NeurIPS 2025) Track on Datasets and Benchmarks.

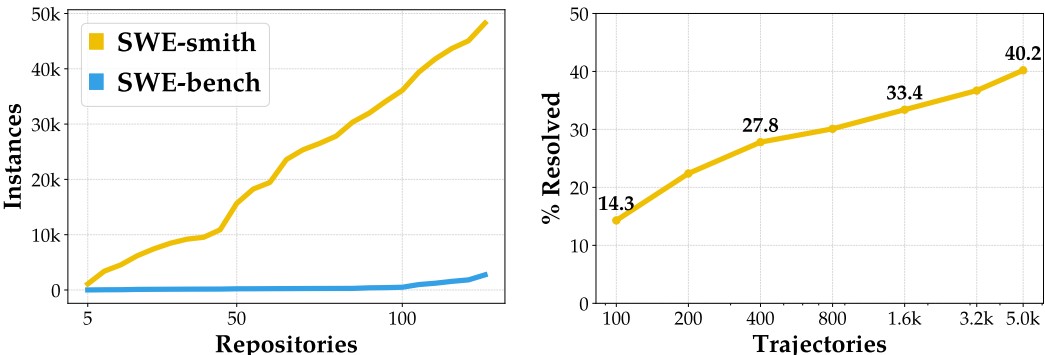

Figure 1: **Scaling task instances** (left) and **performance** (right) for SWE-agent with SWE-smith. Using SWE-smith, we can create 100s to 1000s of instances for any Python codebase, enabling us to train `SWE-agent-LM-32B` which achieves $40.2\%$ on SWE-bench Verified.

In this paper, we introduce the SWE-smith toolkit, which marries the flexible execution environments of SWE-bench with scalable instance collection (Figure 1). SWE-smith features several techniques to automatically synthesize bugs in existing GitHub repositories, such as (1) generating errant rewrites of functions with an LM, (2) procedurally modifying the abstract syntax tree (AST) of functions, (3) undoing PRs, and (4) combining bugs. Our key insight is that execution-based validation can not only validate proposed solutions, but also identify bug candidates which cause substantial software regression (i.e., break tests).

In a nutshell, SWE-smith puts forth the following task creation workflow, as shown in Figure 2. Given a codebase, we automatically set up a corresponding environment using SWE-agent [49]. Within this environment, we then use the aforementioned techniques to synthesize 100s to $1,000$s of task instances. Finally, we craft realistic issue descriptions automatically with LMs. SWE-smith's design significantly reduces the amount of human labor and storage required for constructing execution environments. Using SWE-smith, we create a dataset of 50k task instances across 128 real-world GitHub repositories.

Using the SWE-smith dataset, we achieve a new open-weight state of the art result on SWE-bench verified. Using the SWE-smith task instances, we generate 5,016 expert trajectories with Claude 3.7 Sonnet and fine-tune Qwen 2.5 Coder Instruct 32B. The resulting LM, `SWE-agent-LM-32B`, achieves $40.2\%$ (+33.4%) on SWE-bench Verified in a single attempt, without inference-time scaling. This sets a new state of the art for open-weight models.

The scale and diversity of the SWE-smith dataset enables us to investigate optimal strategies for training SE agents. Training on more instances, bug types, and repositories improves LM performance. LMs can generate realistic issue texts from bug patches. Using SWE-smith, it is possible to optimize LMs for specific repositories while only suffering minor generalization loss.

We release SWE-smith as an open-source toolkit — including instances, environments, and trajectories — to catalyze the development of stronger open-source LM agents.

## 2 SWE-smith: Software Task Generation at Scale

The core principle of SWE-smith's collection strategy is to define an execution environment first, and then synthesize task instances within the environment. Conceptually, this is a simple inversion of SWE-bench's approach, which instead prioritizes identifying task instances, and then attempts to build an environment for each. In this section, we describe the procedure in detail and show how, in practice, SWE-smith scales significantly better in terms of repositories, task instances, and storage.

### 2.1 Collection

**Building execution environments for repositories with passing tests.** Given a repository, we run SWE-agent [49] on the latest commit for at most 100 steps, instructing it to install the codebase and run the test suite. We then manually verify the installation and testing instructions, check if more than 80% of existing tests pass, and finally create a Docker image for the repository. We target repositories

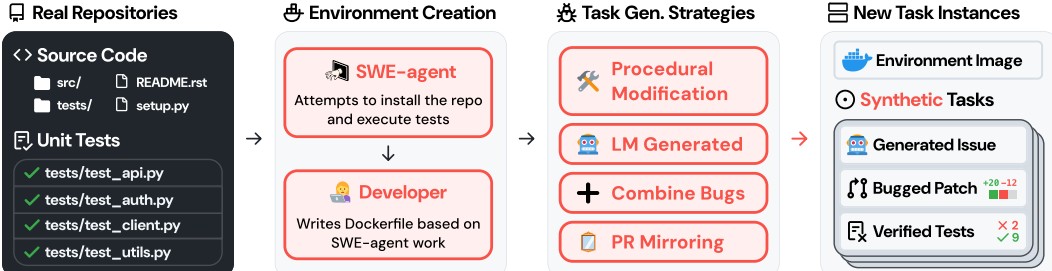

Figure 2: SWE-smith creates training data for software engineering agents by crafting bugs in real codebases. We employ several strategies to create task instances that break existing tests. Using SWE-smith, we create 50k+ task instances with execution environments from 128 real world repositories.

for the 5K most downloaded packages listed in the Python Package Index (PyPI) as of Nov. 18, 2024, then filter out any PyPI package with fewer than 1K stars on GitHub, as well as the 12 SWE-bench test repositories. More in §A.2.

**Creating task instance candidates.** Per repository, we employ four different strategies to create candidates. As shown in Figure 2, each strategy takes in a repository as input, then produces task instance candidates represented as `.diff` files. We provide extensive details in §B.

- **LM Generation**: Per repository, we identify all programmatic entities (functions, classes), then take two approaches: (1) provide an LM with the function and prompt it to introduce errant *modifications* (henceforth referred to as "LM Modify"), and (2) given only the function header and docstring, ask the LM to *rewrite* it ("LM Rewrite"). See more in §B.1.

- **Procedural Modification**: Per function, we acquire an abstract syntax tree (AST) representation of the code, then randomly perform one or more transformations (e.g., remove a conditional/loop, change an operator, and 11 more. See Table 8). See more in §B.2.

- **Combine Bugs**: LM generation and Procedural Modification task instances exclusively edit one function or class. To create more complex tasks that require editing multiple portions of the codebase, we devise a "Patch Combination" strategy that creates a task instance by aggregating candidates from the same file(s) or module(s). See more in §B.3.

- **Invert PRs** (or "PR Mirror"): Per repository, we collect all PRs that modify Python files. Per PR, we attempt to *undo* its revisions in the current version of the repository. To achieve this, we provide an LM with the PR's code changes (a `.diff` plaintext) and prompt it to rewrite each affected file such that the PR edits are reverted. Unlike SWE-bench, we do *not* check out the PR's base commit, as the install specifications determined in the previous step may not be compatible with older versions of the repo. See more in §B.4.

**Execution-based validation of candidates.** We apply each candidate patch to the corresponding repository, run the test suite, and only keep patches that break one or more existing, passing tests (referred to as *Fail-to-Pass* or *F2P* test(s)). For efficiency purposes, we discard bug candidates where test runtimes exceed two minutes. We provide minor additional details in §A.3.

**Generating problem statements.** The issue text associated with a bug can significantly alter the difficulty and feasibility of the task instance. Providing descriptions of "expected" vs. "observed" behavior or reproduction code heavily affects an agent's capacity to localize bugs or iterate on proposed solutions. We explore several techniques covered fully in §C, and ultimately settle on a simple strategy. Per task instance, we provide an LM with the `.diff` patch, source code of a random F2P test, and execution output from running the repository's test suite with the bug patch applied. We prompt the LM to mimic the style of Github issues and to include reproduction code based on the F2P test.

**What human labor remains?** The steps requiring manual effort are (1) parsing the correct installation setup procedures from the agent trajectory ($\sim$ 7 min per repository), and (2) implementing the parser for test outputs ($\sim$ 1 min per repository). Step two requires very little time because parsers can be reused for repositories with the same testing infrastructure (e.g., `pytest`). SWE-smith removes the need for resolving installation issues for multiple versions of a codebase across time, the most costly step of SWE-bench collection. Creating SWE-smith took one author $\sim$ 20h of human labor.

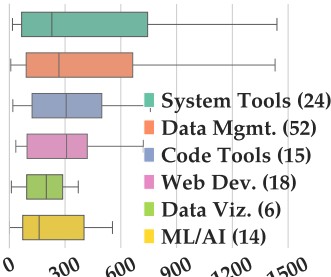

Figure 3: Distribution of instances per repo for 128 repos grouped into 6 categories.

Table 1: Summary of SWE-smith statistics. "Yield %" is the % of candidates generated by a strategy that break 1+ tests. "Cost" is the average cost to generate one candidate. "F2P" (Fail to Pass tests), "Lines [Edited]" are median values.

| Bug Type | Yield % | # Insts | Cost | F2P | Lines |
|---|---|---|---|---|---|
| Combine | 96.9% | 10,092 | 0.00¢ | 15 | 11 |
| LM Modify | 56.0% | 17,887 | 0.38¢ | 4 | 3 |
| LM Rewrite | 35.0% | 4,173 | 3.93¢ | 4 | 24 |
| PR Mirror | 33.8% | 2,344 | 5.53¢ | 3 | 14 |
| Procedural | 40.2% | 15,641 | 0.00¢ | 7 | 5 |
| Total | 50.1 | 50,137 | 2.32¢ | 6 | 5 |

## 2.2 Features

We apply SWE-smith to 128 Python repositories, generating a total of 50k instances. Table 1 captures the key statistics. On average, we generate 381 task instances per repository, with as many as 2277 for `pandas-dev/pandas`. We summarize the distribution of task instances per repository in Figure 3, where repositories are grouped into one of six general categories. SWE-smith took $1360 to create ($1000 to generate bugs, $160 for automatic repository installation with SWE-agent, $200 to generate issues for 10K bugs). Generating an issue costs 2.54¢ on average. More dataset analyses in §D.

Bug generation strategies vary in cost and yield rate. Of methods reliant on LMs, PR Mirrors are most expensive because the task entails rewriting entire files, as opposed to individual functions (LM Modify, LM Rewrite). Yield rates are affected by lack of test coverage for the change, or because the bug candidate did not actually introduce an issue. For LM Rewrite, the LM is simply asked to re-implement the function. When a bug is requested outright (LM Modify), the yield is higher.

**How difficult are SWE-smith task instances?** To determine whether task instances produced by SWE-smith are realistic and challenging, we train a Qwen 2.5 32B model on 1,699 human-annotated (task, label) pairs from Chowdhury et al. [9] to rate tasks as (`easy`, `medium`, `hard`) by training. To quantify difficulty, each difficulty label corresponds to values of 1/5/9. The model achieves 75.3% test accuracy. We then rate difficulty of task instances from both SWE-smith and prior SWE-bench style datasets [9, 18, 31, 50]. SWE-smith task instances span a broad range of difficulties, similar to SWE-bench and SWE-gym. The average difficulty score for SWE-smith (5.27–5.72 across bug generation strategies) is comparable to SWE-bench (5.01) and SWE-gym (5.62). This suggests SWE-smith enables realistic and appropriately challenging evaluation. We discuss why bug strategies yield different levels of difficulty and visualize difficulty per dataset in §E.

**Scaling execution environments.** Unlike SWE-bench which creates a Docker image per task instance, SWE-smith leverages a simpler design where tasks from the same repository share the same environment, reducing storage overhead significantly, as shown in Table 2. This approach reduces

Table 2: Comparison of open source training datasets for software engineering tasks. Relative to existing datasets, SWE-smith has multiple times the number of task instances, repositories, and environments at a fraction of prior storage costs. SWE-fixer and SWE-bench-train task instances do not have execution environments, so "Env. Size" is blank.

| Dataset | # Tasks | # Repos | Exec? | Source | Env. Size |
|---|---|---|---|---|---|
| R2E [15] | 0.25k | 137 | ✓ | Synth | 270 GBs |
| R2E-gym (Subset) [16] | 4.6k | 10 | ✓ | Synth | 4 TBs |
| SWE-bench-extra [5] | 6.38k | 2k | ✗ | Real | - |
| SWE-bench-train [18] | 19k | 37 | ✗ | Real | - |
| SWE-fixer [44] | 115k | 856 | ✗ | Real | - |
| SWE-gym [31] | 2.4k | 11 | ✓ | Real | 6 TBs |
| **SWE-smith** | 50k | 128 | ✓ | Both | 295 GBs |

costs and makes SWE-smith more easily accessible and maintainable than existing datasets. We estimate that creating a similar quantity of task instances (50k) using SWE-bench would require 50 to 150 TBs of storage for environments, a 500x difference. Extended discussion in §D.1.

## 3 Experiments

To explore the utility of SWE-smith for training software engineering agents, we use rejection sampling fine-tuning [54] as the primary procedure for improving a base LM with SWE-smith. Our experiment workflow is as follows. First, we curate a subset of SWE-smith task instances. Next, we run an agent system with an expert model on this subset. At this step, the trajectory corresponding to each run is recorded. Then, we fine-tune the base (or "student") model on the trajectories corresponding to resolved instances. Finally, we evaluate the agent system run with the student model on a separate, test split.

**Models.** For expert models, we use `claude-3-7-sonnet-20250219` [2]. For fair comparisons with prior works [31], we also use `claude-3-5-sonnet-20240620` and `gpt-4o-2024-08-06`. We use the `Qwen-2.5-Coder-Instruct` [14] 7B and 32B series as the base models. Training and hyperparameter details are in §F.1.

**Agent system.** We use SWE-agent [49], an agent system for solving GitHub issues. SWE-agent provides a base LM with an Agent Computer Interface (ACI) that enables more effective interactions with a codebase. At each turn, SWE-agent prompts an LM to generate a ReAct [52] style (thought, action) pair, where the action either edits a file or executes a shell command. We choose SWE-agent because, at the time of writing, SWE-agent with Claude 3.7 Sonnet is the top open source solution on SWE-bench. When generating trajectories with expert models, we run SWE-agent for at most 75 steps and $2.00 cost limit. For inference of student models, we impose the same 75 step maximum and fix temperature at 0.0. Full configuration details are in §F.1.

**Evaluation metrics.** We evaluate on SWE-bench Lite and Verified [9]. SWE-bench evaluates AI systems on their ability to solve software issues from 12 real world GitHub repositories. The Lite split consists of 300 tasks, curated to be an easier evaluation set that's less costly to run. The Verified split is a human-curated subset of 500 instances, selected for clearer problem statements and more reliable evaluation. We report the **% resolved** metric, the proportion of successfully resolved instances.

## 4 Results

Table 3 compares the performance of Qwen 2.5 Coder Instruct models (7B and 32B), fine-tuned on 5,016 SWE-smith trajectories. We refer to them as `SWE-agent-LM-7B` and `SWE-agent-LM-32B`; the latter achieves state-of-the-art performance.

The final dataset of 5,016 training points was curated as follows. We start by collecting a large pool of expert trajectories. First, we carried out each of the ablations in Section 4.1, giving us an initial set of 5,105 trajectories. Next, based on our observation that PR Mirror and LM Rewrite task instances yield the most effective expert trajectories (discussed below), we run the expert model on all task instances of these types, bumping up the total number to 6,457 task instances. Ultimately, we attempt to generate expert trajectories for 8,686 unique task instances, or 17.3% of the SWE-smith dataset. Reinforcing the difficulty rating findings from Section 2.2, we observe that SWE-smith task instances are non-trivial for the top agent systems today. The final pool of 6,457 represents a 36% resolve rate of all 17,906 attempts to solve one of the 8,686 task instances.

Next, we perform minor filtering of this collection. As reported in Pan et al. [31], we also observe that "easier" trajectories – task instances that are repeatedly solved across multiple runs — degrade model performance. Therefore, we limit the number of times any SWE-smith task instance is represented in the training set to 3 trajectories. This leads to the final 5,016 training set. More details in §F.3.

**Performance improves with more data points.** Extending similar scaling graphs from prior works [16, 31], Figure 1 shows increasing performance with more trajectories.

**Comparison at the same training set size.** To compare with prior works [16, 31], we run expert trajectory generation on 1000 random SWE-smith task instances with SWE-agent + Claude 3.5 Sonnet (800) or GPT-4o (200). We then fine-tune the 32B model on 500 successful trajectories, a

Table 3: Resolve rates for existing solutions on SWE-bench Lite and Verified, collected from Jimenez et al. [17], compared to models fine-tuned on SWE-smith. All performance numbers are pass@1. We do *not* compare against systems that use verifiers or multiple attempts at test time.

| Model | System | Train Size | Lite | Verified |
|-------|--------|-----------|------|----------|
| *Closed Weight Models* | | | | |
| GPT-4o [27] | Agentless | - | 32.0 | 38.8 |
| | OpenHands | - | 22.0 | - |
| | SWE-agent | - | 18.3 | 23.0 |
| Claude 3.5 Sonnet [1] | Agentless | - | 40.7 | 50.8 |
| | AutoCodeRover | - | - | 46.2 |
| | OpenHands | - | 41.7 | 53.0 |
| | SWE-agent | - | 23.0 | 33.6 |
| Claude 3.7 Sonnet [2] | SWE-agent | - | **48.0** | **58.2** |
| Llama3-SWE-RL-70B [41] | Agentless | 11M | - | 41.0 |
| *Open Weight Models* | | | | |
| Lingma-SWE-GPT-72B [22] | SWE-SynInfer | - | - | 28.8 |
| Qwen3-235B-A22B [33] | OpenHands | - | - | 34.4 |
| R2E-Gym-32B [16] | OpenHands | 3.3k | - | 34.4 |
| SWE-fixer-72B [44] | SWE-Fixer | 110k | 24.7 | 32.8 |
| SWE-gym-32B [31] | OpenHands | 491 | 15.3 | 20.6 |
| SWE-agent-LM-7B | SWE-agent | 2k | 11.7 | 15.2 |
| SWE-agent-LM-32B | SWE-agent | 5k | **30.7** | **40.2** |

training set size both works report on. Our model achieves a 28.2% resolve rate on SWE-bench Verified, a relative difference of +8.2% with Pan et al. [31] and +0.7% with Jain et al. [16].

## 4.1 Ablations of SWE-smith

We perform several ablations of how SWE-smith's bug and problem statement generation strategies impact the quality of training data. Unless otherwise specified, we use Claude 3.7 Sonnet as the expert for fine-tuning Qwen 2.5 7B Coder Instruct, and report the performance on SWE-bench Verified.

**LM Rewrite and Procedural bugs are comparable to PR mirrors.** We randomly sample 1000 instances per bug generation strategy (LM Modify, LM Rewrite, Procedural Modifications, PR Mirrors). For each instance, we generate issue text with an LM and run expert trajectory generation. We then fine-tune separate student models per strategy, capping training points to the minimum number of successful trajectories from any strategy (507) for fair comparison.

Table 4 summarizes the results. Trajectories generated from PR mirrors are empirically the most effective training data — this is expected, since they are most reflective of SWE-bench. What's noteworthy is that trajectories from Procedural Modification and LM Rewrite instances lead to competitive models. There is a steep drop-off with LM Modify bugs.

**LM generated issues are comparable to real issues.** We randomly sample 600 PR Mirror task instances. We compare LM generated issues with three alternatives — fixed issue templates, the source code + test logs of a random Fail-to-Pass test, and the original issue text associated with the PR. We again cap training points to the minimum number of successful trajectories (259) for fairness.

As shown in Table 5, training on task instances with LM generated issues is empirically comparable to using the original issue text. Using fixed issue templates not only leads to the fewest successful trajectories, but also results in relatively homogeneous problem solving sequences. The expert trajectories from fixed issue templates have 31% fewer unique actions compared to LM generated text (379 vs. 550). While providing a Fail-to-Pass test case leads to more successful expert trajectories, leaking the evaluation criteria causes the model to skip over writing a reproduction script, which accounts for the performance drop. Of 500 SWE-bench Verified instances, the student model trained on LM-generated issues attempts to reproduce the bug for 379 of the runs. The model trained on test-based issues only does so for 127 cases, a 66% decrease.

Table 4: Comparison of training on 1000 SWE-smith instances created with different strategies.

| Strategy | # Trajs. | % Resolved |
|---|---|---|
| LM Modify | 802 | 5.7 ($\pm$1.5) |
| LM Rewrite | 507 | 8.8 ($\pm$1.7) |
| Procedural | 745 | 8.6 ($\pm$1.8) |
| PR Mirror | 557 | 9.2 ($\pm$1.7) |

Table 5: Comparing training on 600 PR Mirror instances with varied issue text.

| Issue | # Trajs. | % Resolved |
|---|---|---|
| Fixed | 259 | 6.4 ($\pm$1.5) |
| F2P Test | 390 | 7.3 ($\pm$1.9) |
| LM | 328 | 7.7 ($\pm$1.5) |
| Original | 319 | 7.8 ($\pm$1.8) |

**Task difficulty correlates with solvability but not with effectiveness as training data.** First, we run our difficulty rating model on 10k randomly selected SWE-smith task instances. From this pool, we curate subsets of 1000 instances corresponding to the three difficulty levels, then run expert trajectory generation per subset 3 times. For the `easy/medium/hard` subsets, the resolve rate by the expert model are 58.6%, 41.0%, and 17.0% respectively.

Next, from all successful trajectories, we create four fine-tuning datasets of 500 trajectories each corresponding to difficulty scores of 2, 4, 6, and 8. As mentioned in Section 2.2, the corresponding scores for `easy/medium/hard` are 1/5/9. Therefore, the SFT dataset for score 2 is made up of trajectories corresponding to 375 `easy` and 125 `medium` instances, and so on. Somewhat surprisingly, we do not observe strong correlation between increased difficulty and downstream performance. For the student models trained on the 2/4/6/8 difficulty SFT datasets, we get pass@1 scores of 12.4%, 10.8%, 13.6%, and 12.2% on SWE-bench Verified.

**Training on more repositories improves general performance.** We train models in four settings by sampling 700 expert trajectories on Procedural Modification tasks from pools of 4, 25, 50, and 100 repositories. Echoing similar findings for code generation tasks [46], we find that increasing repositories represented in the training set improves performance, as shown in Figure 5, with an approximately logarithmic relation between model performance and number of repositories.

**Repository-specialized models excel on the target repository with minor generalization loss.** We experiment with training models to be specialists on one particular repository. To assess performance, we evaluate models on a subset of SWE-bench Verified tasks that are (1) from SymPy, and (2) created after January 1st, 2022, a total of 22 instances. To create SymPy specific training data, we first select a base commit of SymPy just before the cutoff date. Next, we create 1276 Procedural Modification task instances, then generate 700 expert trajectories. We evaluate specialization in two settings: (1) single-repository fine-tuning, and (2) specialist stage fine-tuning, both shown in Figure 4. For single-repository tuning, we compare a model initialized with `Qwen-2.5-Coder-Instruct` 7B and trained on 700 instances sampled from 100 repositories, to the same Qwen base model but fine-tuned on the 700 SymPy instances only. For specialist stage fine-tuning, we simply compare `SWE-agent-LM-32B` to the same model further fine-tuned on the 700 SymPy instances.

Specialization significantly boosts performance for the target repository with only slight drops in general performance in both the single-repository fine-tuning (21.2% vs. 13.6%) and specialist stage fine-tuning (42.4% vs. 33.3%) settings, compared to baselines trained across 128 repositories.

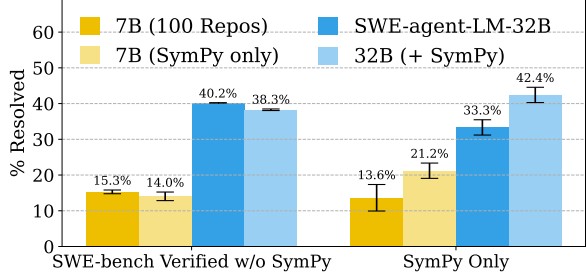

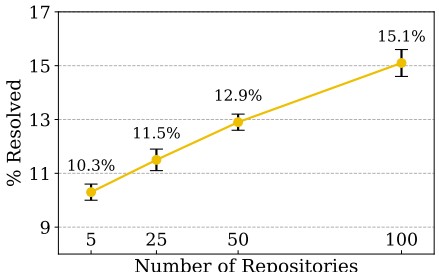

Figure 4: We fine-tune a 7B base and our 32B models on 700 trajectories for SymPy. Specialization boosts SymPy performance, with minimal generalization loss.

Figure 5: At 700 training samples, we observe performance increases logarithmically with number of repositories.

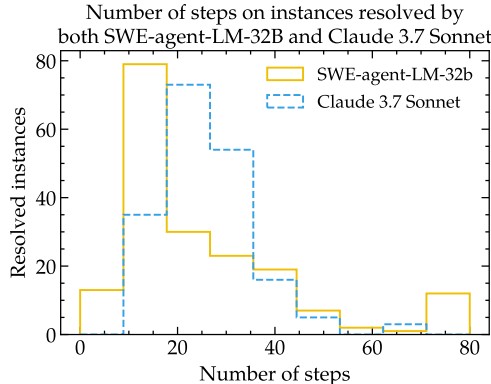

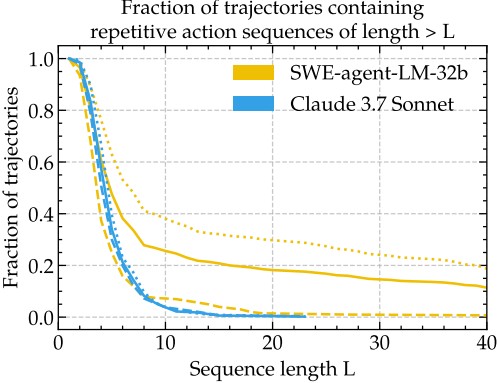

Figure 6: `SWE-agent-LM-32B` takes fewer steps to submit a solution compared to Claude 3.7 Sonnet for task instances resolved by both models.

Figure 7: For unsuccessfully resolved tasks, a frequent failure mode is that `SWE-agent-LM-32B` will repeat actions.

## 4.2 Analysis of Agent Behavior

This section analyzes the behavior, failure modes, and efficiency of SWE-agent when run with `SWE-agent-LM-32B` or Claude 3.7 Sonnet on SWE-bench verified.

**SWE-agent-LM-32B can solve tasks efficiently.** `SWE-agent-LM-32B` resolves tasks in fewer steps on average (24.9) than Claude 3.7 Sonnet (29.1), though the difference becomes marginal when accounting for different average difficulties of the resolved tasks: On the overlap of tasks that are resolved by both LMs, `SWE-agent-LM-32B` uses 24.8 steps compared to 25.6 used by Claude 3.7 Sonnet (see Fig. 6). While shorter trajectories are not always preferred (additional actions can be used for additional validation purposes, for example), this shows that `SWE-agent-LM-32B` solves tasks very efficiently. At the same time `SWE-agent-LM-32B` also demonstrates that it can remain focused throughout long trajectories, with 31 instances being resolved after 40 steps or more. We further highlight that the accuracy of naturally terminating [1] agent submissions with `SWE-agent-LM-32B` achieve an accuracy nearly matching that of Claude 3.7 Sonnet (60% vs 63%), showing that `SWE-agent-LM-32B` is adept at determining whether an instance has been resolved. As the overall cost and turn count averages scale strongly with the cost and turn limits, we reserve a more thorough analysis for §F.5.1.

**Repetitive actions are a key problem.** We observe a tendency for `SWE-agent-LM-32B` to get stuck in long sequences of repetitive actions, in particular long sequences of calls that display different portions of a file instead of using search commands. [2] More than 25% of `SWE-agent-LM-32B` trajectories have a repetitive sequence of at least length 10, compared to less than 4% for Claude 3.7 Sonnet (see Figure 7). The occurrence of long repetitive sequences correlates strongly with the agent's ability to solve the corresponding task instance, largely because the LM continues issuing similar commands until either the agent cost or turn limit is reached, at which point the run is terminated. For example, repetitive sequences of length 10 correspond to an 89% failure probability. Simple interventions from the agent scaffold can mitigate repetitive actions, but do not seem to improve resolve rates (see §F.5).

**Localization is the dominant failure mode.** Guided by a short plan in the system prompt, SWE-agent typically starts by *localizing* (search and read actions), *reproducing* (test file creation and execution), before modifying source files and validating the fixes. If the agent gets stuck at any of these stages or keeps on iterating, the agent loop is eventually interrupted by runtime limits (cost, number of LM calls, runtime). While this rarely happens with Claude 3.7 Sonnet, 53% of `SWE-agent-LM-32b`'s failures are associated with such limits (Figure 8). The agent often already gets stuck during localization or initial efforts to reproduce a bug, with endlessly repeated actions being a persistent issue. More on failure modes in §F.5.

---

[1] i.e., excluding agent runs that are terminated due to errors or cost/step count limits. Note that SWE-agent still extracts and submits any changes performed by the agent in these cases and some of them can be successful (for example if the agent is terminated due to cost while testing already performed edits).

[2] In fact, these `str_replace_editor view` commands make up 73% of the longest repetitive sequences. For this analysis, we look at repetitions of the base command, i.e., without any arguments. See §F.5 for more.

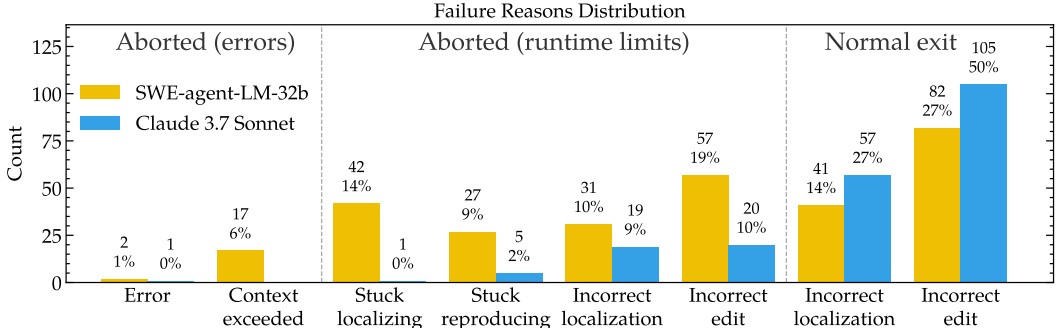

Figure 8: More than half of the unresolved instances of `SWE-agent-LM-32B` correspond to runs terminated by cost/step limits, and these limits are frequently reached before source code has been modified. See §F.5 for more.

## 5 Related Work

**LMs for Software Engineering.** As contemporary LMs have saturated traditional code generation tasks [4, 8], software engineering benchmarks [15, 18, 50, 58, 55], notably SWE-bench, have become a new de facto evaluation setting due to their diverse, complex, real-world programming challenges. The most significant source of open source progress on SWE-bench has been the development of LM-based workflows [29, 42, 57] and agents [3, 38, 49, 56]. Workflow-based systems are typically human-engineered decompositions of a task into a sequence of sub-goals. Yang et al. [50] suggests such pipelines may not generalize effectively to non-Python repositories, requiring additional human intervention to re-adapt. We therefore elect to focus on generating trajectories with and for LM agent systems [36, 48, 52]. Because no workflow is imposed, agent systems inherently rely more on the LM to plan and refine its actions, putting more focus on an LM's capabilities, not inference scaffolds.

**Training Datasets for Coding.** Prior work around training data has focused on instruction following [21, 24, 35, 39, 40, 53] and preference learning [19, 20] for code completion tasks. Several recent works introduce training sets for retrieval augmented generation [18, 44], workflows [41], and agent [5, 22, 31, 16] approaches to SWE-bench. Our work applies Haluptzok et al. [12] at a repository level: by having an LM break a codebase, we drastically reduce the human effort needed to define a task and build its environment. Concurrent to our work, Xie et al. [46] (RePOST) also constructs execution environments for repository functions, but differs significantly in methodology and evaluation. RePOST sandboxes a function and its dependencies to a separate script, then generates tests with an LM, removing the original codebase as context. The tasks' source is repository-level; the environments and tasks are not. RePOST evaluates solely on code generation (e.g., HumanEval [8]). Jain et al. [16] (R2E-Gym) improves open source LMs' performance on SWE-bench with inference time scaling and verifiers. R2E-gym's 51% resolve rate is not comparable to Table 3 results, as each instance is attempted 26 times. R2E-gym's 4.6k training instances are collected using SWE-bench's pipeline, with some augmentations around using LMs to synthesize issue text and tests. To our knowledge, we are the first to address the limited scalability of previous approaches.

## 6 Discussion

**Limitations and future directions.** First, SWE-smith's collection pipeline is Python-centric. The mechanisms to identify programmatic objects and perform transformations rely heavily on the Python specific `ast` library. That said, SWE-smith's collection strategy is transferable to other languages. Second, due to both compute constraints and our work's primary focus on contributing a dataset, we only explore fine-tuning to demonstrate SWE-smith's effectiveness. Future work could explore other training techniques such as eliciting agentic capabilities via reinforcement learning.

**Conclusion.** We introduce SWE-smith, a dataset of 50k software engineering task instances from 128 real world GitHub repositories. SWE-smith collection pipeline scales up task instances, environments, and trajectories at a fraction of prior costs without sacrificing faithfulness to open source software development practices. Using SWE-smith, we train `SWE-agent-LM-32B`, achieving a state-of-the-art 40.2% on SWE-bench Verified. Our experiments show how SWE-smith leads to key insights on how to develop SWE-agents. We believe SWE-smith provides the foundational infrastructure needed to train software engineering agents in a truly scalable manner.

## Acknowledgments and Disclosure of Funding

We thank Princeton Language & Intelligence (PLI) for providing credits for running closed-source API models. Thanks to Samuel Ainsworth for his constant support of `bitbop.io` (`https://bitbop.io/`), the compute service for which the majority of the project was carried out with. We'd also like to thank Akshat Bubna, Howard Halim, Andrew Liu, Peyton Walters, and the great team at Modal (`https://modal.com/`) for providing credits that made fine-tuning and model serving efforts extremely easy for this project. This work is partially supported by ONR grant N000142412532 and NSF grant IIS-2247357. We also thank Open Philanthropy and Andreessen Horowitz for providing funding for this work. Finally, thanks to Tianyu Gao, William Held, Niklas Muennighoff, Rafael Rafailov, Yijia Shao, Chenglei Si, Anikait Singh, Tianyi Zhang, Kexin Pei, and Karthik Narasimhan for constructive discussions and support throughout this project.

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

# NeurIPS Paper Checklist

1. **Claims**

   Question: Do the main claims made in the abstract and introduction accurately reflect the paper's contributions and scope?

   Answer: [Yes]

   Justification: In the abstract and introduction (Section 1), we highlight the size of our SWE-smith dataset (50k task instances collected from $128$ repositories), preview several collection strategies, and highlight `SWE-agent-LM-32B` which achieves state of the art performance on SWE-bench Verified. Our paper is organized such that each claim is discussed. In Section 2, we discuss the collection strategies and characterize the dataset. In Section 4, we discuss the main result, and also provide many additional ablations reinforcing our findings.

   Guidelines:

   - The answer NA means that the abstract and introduction do not include the claims made in the paper.
   - The abstract and/or introduction should clearly state the claims made, including the contributions made in the paper and important assumptions and limitations. A No or NA answer to this question will not be perceived well by the reviewers.
   - The claims made should match theoretical and experimental results, and reflect how much the results can be expected to generalize to other settings.
   - It is fine to include aspirational goals as motivation as long as it is clear that these goals are not attained by the paper.

2. **Limitations**

   Question: Does the paper discuss the limitations of the work performed by the authors?

   Answer: [Yes] .

   Justification: We mention limitations inline for different bug generation methods in Section 2, and also have a dedicated paragraph for covering the most glaring shortcomings in Section 6. We have much more extensive discussions of what can be improved upon and offer concrete future research directions in §B for different bug generation techniques and §F for training methods that we can explore. We also curate a SWE-bench Multilingual dataset, explicitly demonstrating that the current SFT approach doesn't really encourage generalizability, an extremely actionable next step that SWE-smith can readily be used for.

   Guidelines:

   - The answer NA means that the paper has no limitation while the answer No means that the paper has limitations, but those are not discussed in the paper.
   - The authors are encouraged to create a separate "Limitations" section in their paper.
   - The paper should point out any strong assumptions and how robust the results are to violations of these assumptions (e.g., independence assumptions, noiseless settings, model well-specification, asymptotic approximations only holding locally). The authors should reflect on how these assumptions might be violated in practice and what the implications would be.
   - The authors should reflect on the scope of the claims made, e.g., if the approach was only tested on a few datasets or with a few runs. In general, empirical results often depend on implicit assumptions, which should be articulated.
   - The authors should reflect on the factors that influence the performance of the approach. For example, a facial recognition algorithm may perform poorly when image resolution is low or images are taken in low lighting. Or a speech-to-text system might not be used reliably to provide closed captions for online lectures because it fails to handle technical jargon.
   - The authors should discuss the computational efficiency of the proposed algorithms and how they scale with dataset size.
   - If applicable, the authors should discuss possible limitations of their approach to address problems of privacy and fairness.

- While the authors might fear that complete honesty about limitations might be used by reviewers as grounds for rejection, a worse outcome might be that reviewers discover limitations that aren't acknowledged in the paper. The authors should use their best judgment and recognize that individual actions in favor of transparency play an important role in developing norms that preserve the integrity of the community. Reviewers will be specifically instructed to not penalize honesty concerning limitations.

3. **Theory assumptions and proofs**

   Question: For each theoretical result, does the paper provide the full set of assumptions and a complete (and correct) proof?

   Answer: [NA]

   Justification: This work is not particularly theoretical. It is mainly a dataset contribution, and is meant to encourage more focus in a rather empirically driven and applied area of Language Model development. Therefore, this is not applicable.

   Guidelines:

   - The answer NA means that the paper does not include theoretical results.
   - All the theorems, formulas, and proofs in the paper should be numbered and cross-referenced.
   - All assumptions should be clearly stated or referenced in the statement of any theorems.
   - The proofs can either appear in the main paper or the supplemental material, but if they appear in the supplemental material, the authors are encouraged to provide a short proof sketch to provide intuition.
   - Inversely, any informal proof provided in the core of the paper should be complemented by formal proofs provided in appendix or supplemental material.
   - Theorems and Lemmas that the proof relies upon should be properly referenced.

4. **Experimental result reproducibility**

   Question: Does the paper fully disclose all the information needed to reproduce the main experimental results of the paper to the extent that it affects the main claims and/or conclusions of the paper (regardless of whether the code and data are provided or not)?

   Answer: [Yes]

   Justification: We provide detailed descriptions of the experimental setup, including the models used (e.g., Claude 3.7 Sonnet, Qwen 2.5 Coder Instruct), training procedures, and evaluation metrics in Section 3 and Appendix F. Additionally, we describe the dataset creation process in Section 2 and provide open access to the dataset, code, and models via `https://swesmith.com/`. The supplemental material includes instructions for reproducing the results, such as hyperparameters, data splits, and compute requirements. These details ensure that the main experimental results can be faithfully reproduced.

   Guidelines:

   - The answer NA means that the paper does not include experiments.
   - If the paper includes experiments, a No answer to this question will not be perceived well by the reviewers: Making the paper reproducible is important, regardless of whether the code and data are provided or not.
   - If the contribution is a dataset and/or model, the authors should describe the steps taken to make their results reproducible or verifiable.
   - Depending on the contribution, reproducibility can be accomplished in various ways. For example, if the contribution is a novel architecture, describing the architecture fully might suffice, or if the contribution is a specific model and empirical evaluation, it may be necessary to either make it possible for others to replicate the model with the same dataset, or provide access to the model. In general. releasing code and data is often one good way to accomplish this, but reproducibility can also be provided via detailed instructions for how to replicate the results, access to a hosted model (e.g., in the case of a large language model), releasing of a model checkpoint, or other means that are appropriate to the research performed.

- While NeurIPS does not require releasing code, the conference does require all submissions to provide some reasonable avenue for reproducibility, which may depend on the nature of the contribution. For example
    (a) If the contribution is primarily a new algorithm, the paper should make it clear how to reproduce that algorithm.
    (b) If the contribution is primarily a new model architecture, the paper should describe the architecture clearly and fully.
    (c) If the contribution is a new model (e.g., a large language model), then there should either be a way to access this model for reproducing the results or a way to reproduce the model (e.g., with an open-source dataset or instructions for how to construct the dataset).
    (d) We recognize that reproducibility may be tricky in some cases, in which case authors are welcome to describe the particular way they provide for reproducibility. In the case of closed-source models, it may be that access to the model is limited in some way (e.g., to registered users), but it should be possible for other researchers to have some path to reproducing or verifying the results.

5. **Open access to data and code**

   Question: Does the paper provide open access to the data and code, with sufficient instructions to faithfully reproduce the main experimental results, as described in supplemental material?

   Answer: [Yes]

   Justification: As answered in the previous question, we provide open access to the data and code, accessible at `https://swesmith.com/`. We also provide extensive documentation discussing how to use the code at `https://swesmith.com/getting_started/`, which also includes thorough details on how to recreate the main results and ablations. This links are included in the last line of the abstract, and we cover specific training details in both Section 3 and §F.

   Guidelines:

   - The answer NA means that paper does not include experiments requiring code.
   - Please see the NeurIPS code and data submission guidelines (`https://nips.cc/public/guides/CodeSubmissionPolicy`) for more details.
   - While we encourage the release of code and data, we understand that this might not be possible, so "No" is an acceptable answer. Papers cannot be rejected simply for not including code, unless this is central to the contribution (e.g., for a new open-source benchmark).
   - The instructions should contain the exact command and environment needed to run to reproduce the results. See the NeurIPS code and data submission guidelines (`https://nips.cc/public/guides/CodeSubmissionPolicy`) for more details.
   - The authors should provide instructions on data access and preparation, including how to access the raw data, preprocessed data, intermediate data, and generated data, etc.
   - The authors should provide scripts to reproduce all experimental results for the new proposed method and baselines. If only a subset of experiments are reproducible, they should state which ones are omitted from the script and why.
   - At submission time, to preserve anonymity, the authors should release anonymized versions (if applicable).
   - Providing as much information as possible in supplemental material (appended to the paper) is recommended, but including URLs to data and code is permitted.

6. **Experimental setting/details**

   Question: Does the paper specify all the training and test details (e.g., data splits, hyperparameters, how they were chosen, type of optimizer, etc.) necessary to understand the results?

   Answer: [Yes]

   Justification: Yes, we provide these details in Section 3 and §F, where we discuss the hyperparameter settings we used to fine-tune `SWE-agent-LM-32B`, and we also list out the

specific settings used for SWE-agent, the primary inference scaffold we run our experiments with. We also write down every third party service we use for compute (primarily Modal).

Guidelines:

- The answer NA means that the paper does not include experiments.
- The experimental setting should be presented in the core of the paper to a level of detail that is necessary to appreciate the results and make sense of them.
- The full details can be provided either with the code, in appendix, or as supplemental material.

7. **Experiment statistical significance**

Question: Does the paper report error bars suitably and correctly defined or other appropriate information about the statistical significance of the experiments?

Answer: [Yes]

Justification: Yes, for all of our experiments and ablations across Section 3 and §F, we primarily use the Pass@1 statistic to account for variance and report metrics that are reproducible. Each ablation includes written or visualized error bars.

Guidelines:

- The answer NA means that the paper does not include experiments.
- The authors should answer "Yes" if the results are accompanied by error bars, confidence intervals, or statistical significance tests, at least for the experiments that support the main claims of the paper.
- The factors of variability that the error bars are capturing should be clearly stated (for example, train/test split, initialization, random drawing of some parameter, or overall run with given experimental conditions).
- The method for calculating the error bars should be explained (closed form formula, call to a library function, bootstrap, etc.)
- The assumptions made should be given (e.g., Normally distributed errors).
- It should be clear whether the error bar is the standard deviation or the standard error of the mean.
- It is OK to report 1-sigma error bars, but one should state it. The authors should preferably report a 2-sigma error bar than state that they have a 96% CI, if the hypothesis of Normality of errors is not verified.
- For asymmetric distributions, the authors should be careful not to show in tables or figures symmetric error bars that would yield results that are out of range (e.g. negative error rates).
- If error bars are reported in tables or plots, The authors should explain in the text how they were calculated and reference the corresponding figures or tables in the text.

8. **Experiments compute resources**

Question: For each experiment, does the paper provide sufficient information on the computer resources (type of compute workers, memory, time of execution) needed to reproduce the experiments?

Answer: [Yes]

Justification: We highlight the compute resources we used explicitly in §F.1, with additional details in the rest of §F. We list the number of GPUs/nodes we use for fine-tuning, along with additional specifications that should communicate fully what kind of compute resources are necessary to reproduce our paper's experiments in their entirety. There is no hidden, additional compute requirement that is not disclosed in the paper.

Guidelines:

- The answer NA means that the paper does not include experiments.
- The paper should indicate the type of compute workers CPU or GPU, internal cluster, or cloud provider, including relevant memory and storage.
- The paper should provide the amount of compute required for each of the individual experimental runs as well as estimate the total compute.

- The paper should disclose whether the full research project required more compute than the experiments reported in the paper (e.g., preliminary or failed experiments that didn't make it into the paper).

9. **Code of ethics**

Question: Does the research conducted in the paper conform, in every respect, with the NeurIPS Code of Ethics https://neurips.cc/public/EthicsGuidelines?

Answer: [Yes]

Justification: We have reviewed the code of ethics and can confirm that our work is not in violation. Our work does not involve engagement with human participants, so there are no concerns that would be raised around preserving anonymity or personally identifiable information. The collection procedures and focus of the SWE-smith dataset do not have any potential for harm towards humans. We also make sure to respect copyright laws, only sourcing from open source repositories that have explicitly given permission for public use, as shown in Table 6.

Guidelines:

- The answer NA means that the authors have not reviewed the NeurIPS Code of Ethics.
- If the authors answer No, they should explain the special circumstances that require a deviation from the Code of Ethics.
- The authors should make sure to preserve anonymity (e.g., if there is a special consideration due to laws or regulations in their jurisdiction).

10. **Broader impacts**

Question: Does the paper discuss both potential positive societal impacts and negative societal impacts of the work performed?

Answer: [Yes]

Justification: We address this directly in §G, where we talk about how the automation of software might bring about concerns about how bug generation tools can be used maliciously, but ultimately demonstrate how our infrastructure is built in a manner that is mindful of the open source development community. Since our work is in the open, the community effort we hope to grow around SWE-smith will also help mitigate concerns.

Guidelines:

- The answer NA means that there is no societal impact of the work performed.
- If the authors answer NA or No, they should explain why their work has no societal impact or why the paper does not address societal impact.
- Examples of negative societal impacts include potential malicious or unintended uses (e.g., disinformation, generating fake profiles, surveillance), fairness considerations (e.g., deployment of technologies that could make decisions that unfairly impact specific groups), privacy considerations, and security considerations.
- The conference expects that many papers will be foundational research and not tied to particular applications, let alone deployments. However, if there is a direct path to any negative applications, the authors should point it out. For example, it is legitimate to point out that an improvement in the quality of generative models could be used to generate deepfakes for disinformation. On the other hand, it is not needed to point out that a generic algorithm for optimizing neural networks could enable people to train models that generate Deepfakes faster.
- The authors should consider possible harms that could arise when the technology is being used as intended and functioning correctly, harms that could arise when the technology is being used as intended but gives incorrect results, and harms following from (intentional or unintentional) misuse of the technology.
- If there are negative societal impacts, the authors could also discuss possible mitigation strategies (e.g., gated release of models, providing defenses in addition to attacks, mechanisms for monitoring misuse, mechanisms to monitor how a system learns from feedback over time, improving the efficiency and accessibility of ML).

11. **Safeguards**

Question: Does the paper describe safeguards that have been put in place for responsible release of data or models that have a high risk for misuse (e.g., pretrained language models, image generators, or scraped datasets)?

Answer: [Yes]

Justification: The dataset artifacts included with this work do not pose any immediate risks. We anticipate and wrote our system to mitigate two kinds of misuse. First, we designed our bug generation pipeline to only work for mirrors of real repositories, as opposed to on the real repositories themselves. This is to avoid interfering or obstructing the work of the repository's maintainers. The actual training dataset itself does not have any safety risks - no information about humans is collected.

Guidelines:

- The answer NA means that the paper poses no such risks.
- Released models that have a high risk for misuse or dual-use should be released with necessary safeguards to allow for controlled use of the model, for example by requiring that users adhere to usage guidelines or restrictions to access the model or implementing safety filters.
- Datasets that have been scraped from the Internet could pose safety risks. The authors should describe how they avoided releasing unsafe images.
- We recognize that providing effective safeguards is challenging, and many papers do not require this, but we encourage authors to take this into account and make a best faith effort.

12. **Licenses for existing assets**

Question: Are the creators or original owners of assets (e.g., code, data, models), used in the paper, properly credited and are the license and terms of use explicitly mentioned and properly respected?

Answer: [Yes]

Justification: The main kinds of assets we provide attribution for are the GitHub repositories represented in the training dataset produced by SWE-smith, which we list exhaustively in Table 6. We also provide attributions to prior works that we build upon, both in the paper and also in the codebase as well whenever code is adopted or copied.

Guidelines:

- The answer NA means that the paper does not use existing assets.
- The authors should cite the original paper that produced the code package or dataset.
- The authors should state which version of the asset is used and, if possible, include a URL.
- The name of the license (e.g., CC-BY 4.0) should be included for each asset.
- For scraped data from a particular source (e.g., website), the copyright and terms of service of that source should be provided.
- If assets are released, the license, copyright information, and terms of use in the package should be provided. For popular datasets, `paperswithcode.com/datasets` has curated licenses for some datasets. Their licensing guide can help determine the license of a dataset.
- For existing datasets that are re-packaged, both the original license and the license of the derived asset (if it has changed) should be provided.
- If this information is not available online, the authors are encouraged to reach out to the asset's creators.

13. **New assets**

Question: Are new assets introduced in the paper well documented and is the documentation provided alongside the assets?

Answer: [Yes]

Justification: Yes, we have open-sourced every aspect of SWE-smith. The website (`https://swesmith.com/`) has links to every component of SWE-smith. The code, which features

the functionality for generating bugs and training code, is available at `https://github.com/SWE-bench/SWE-smith`. The dataset and models for this work are available at `https://huggingface.co/SWE-bench`. Extensive documentation is provided to describe how the assets should be used at `https://swesmith.com/getting_started/`.

Guidelines:

- The answer NA means that the paper does not release new assets.
- Researchers should communicate the details of the dataset/code/model as part of their submissions via structured templates. This includes details about training, license, limitations, etc.
- The paper should discuss whether and how consent was obtained from people whose asset is used.
- At submission time, remember to anonymize your assets (if applicable). You can either create an anonymized URL or include an anonymized zip file.

14. **Crowdsourcing and research with human subjects**

Question: For crowdsourcing experiments and research with human subjects, does the paper include the full text of instructions given to participants and screenshots, if applicable, as well as details about compensation (if any)?

Answer: [NA]

Justification: Our work does not involve experiments or research with human subjects.

Guidelines:

- The answer NA means that the paper does not involve crowdsourcing nor research with human subjects.
- Including this information in the supplemental material is fine, but if the main contribution of the paper involves human subjects, then as much detail as possible should be included in the main paper.
- According to the NeurIPS Code of Ethics, workers involved in data collection, curation, or other labor should be paid at least the minimum wage in the country of the data collector.

15. **Institutional review board (IRB) approvals or equivalent for research with human subjects**

Question: Does the paper describe potential risks incurred by study participants, whether such risks were disclosed to the subjects, and whether Institutional Review Board (IRB) approvals (or an equivalent approval/review based on the requirements of your country or institution) were obtained?

Answer: [NA]

Justification: Our work does not involve experiments or research with human subjects.

Guidelines:

- The answer NA means that the paper does not involve crowdsourcing nor research with human subjects.
- Depending on the country in which research is conducted, IRB approval (or equivalent) may be required for any human subjects research. If you obtained IRB approval, you should clearly state this in the paper.
- We recognize that the procedures for this may vary significantly between institutions and locations, and we expect authors to adhere to the NeurIPS Code of Ethics and the guidelines for their institution.
- For initial submissions, do not include any information that would break anonymity (if applicable), such as the institution conducting the review.

16. **Declaration of LLM usage**

Question: Does the paper describe the usage of LLMs if it is an important, original, or non-standard component of the core methods in this research? Note that if the LLM is used only for writing, editing, or formatting purposes and does not impact the core methodology, scientific rigorousness, or originality of the research, declaration is not required.

Answer: [Yes]

Justification: Yes, a significant aspect of our project involves the use of language models to generate bugs. We explicitly state this in Section 2, discussing exactly how language models are used, and how they are prompted for this purpose. In §B.1, we also provide the actual prompts verbatim that were used to generate both candidates, ultimately represented in the SWE-smith training, data set.

Guidelines:

- The answer NA means that the core method development in this research does not involve LLMs as any important, original, or non-standard components.
- Please refer to our LLM policy (`https://neurips.cc/Conferences/2025/LLM`) for what should or should not be described.

# Appendix

The appendix is generally structured as follows. In Sections A to D, we review details about SWE-smith's infrastructure and collection strategies for curating the SWE-smith task instances and execution environments, providing comparisons to existing datasets such as SWE-bench and SWE-gym along the way. In Sections E and onward, we discuss more about how we created the trajectories dataset, then provide additional ablations and results showcasing the effectiveness of SWE-smith as a dataset.

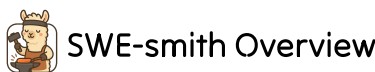

## SWE-smith Overview

### 1. Given a GitHub repository, turn it into an execution environment

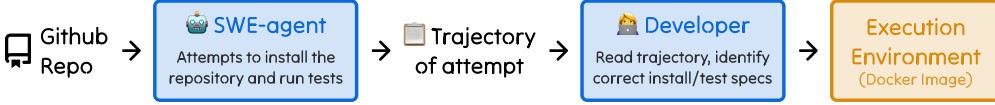

### 2. Given an execution environment, synthesize task instances

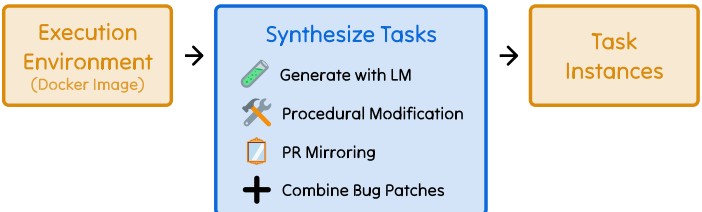

### 3. Given an execution environment + task instances, train SWE-agents!

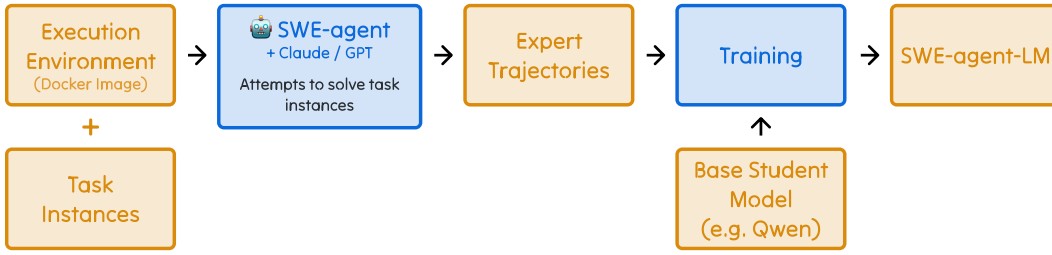

Figure 9: An overview of pipelines in SWE-smith. Scripts/functions and manual steps are highlighted in blue. Artifacts that are also the inputs and outputs of these scripts are in orange. SWE-smith fits in seamlessly with the SWE-bench and SWE-agent ecosystem. Use SWE-smith to construct execution environments and generate task instances. Use SWE-agent to generate expert trajectories on SWE-smith task instances and run inference with models trained on these trajectories. Use SWE-bench to evaluate your models on resolving GitHub issues and performing SWE tasks.

## A   Infrastructure

We cover additional details about how SWE-smith works, specifically

- The form factor of a SWE-smith task instance.

- How we identify repositories and the SWE-agent configuration we use to automatically install them.

- How the task validation and evaluation harnesses work.

### A.1 SWE-smith Task Instance

We briefly review the format of a SWE-smith task instance, highlight how it is different from a SWE-bench task instance, and discuss why SWE-smith's relatively simple infrastructure compared to SWE-bench allows us scale task collection much more efficiently.

A SWE-smith task instance is very similar to the form factor of a SWE-bench task instance, with several minor differences. A SWE-smith task instance includes the following fields:

- `repo`: The repository the task instance is from.
- `instance_id`: A unique identifier (usually `(repo).(bug_type).(hash)`)
- `base_commit`: Hash of the GitHub branch that points to the repository with the bug `patch` applied.
- `patch`: The `diff` that causes the bug. It is applied to the original codebase to create the bug. Reverting this patch is effectively the solution.
- `problem_statement`: The generated issue text that conveys the bug. It is provided to a model or system before it begins attempting a fix.
- `created_at`: A timestamp matching when the bug was successfully validated and pushed to the mirror repository as a branch.
- `FAIL_TO_PASS`: The unit tests that break when the test suite is run with the bug `patch` applied.
- `PASS_TO_PASS`: The unit tests that do not break. These correspond to the set of all tests minus the `FAIL_TO_PASS` tests.

We summarize the key distinctions between a SWE-smith and SWE-bench task instance:

- SWE-smith task instances do not include the `version` or `environment_setup_commit` fields, which SWE-bench requires as additional identifiers for specifying repository-specific installation instructions across time. In SWE-smith, unique installation instructions are specified for each (repository, commit).
- The `hints_text` field is not included. In SWE-bench, this refers to the issue and PR thread comments written after the first commit of the corresponding PR.
- The `created_at` field is assigned the timestamp reflecting when the bug was successfully validated. Originally, `created_at` refers to when a PR was created.
- There is no `test_patch` field, as the SWE-smith collection pipeline does not create or synthesize any hidden tests. All `FAIL_TO_PASS` bugs are visible and runnable in the repository at inference time.

### A.2 Repository Selection

In addition to the criteria discussed in Section 2.1, we also ensure that a repository has a license that allows non-proprietary use. The majority of software licenses are permissive (BSD, MIT, Apache), while the remainder are largely protective licenses (GPL) that still allow for non-commercial use. We inspected the repositories with custom licenses and confirmed they allowed for the use cases exercised in our work. The licenses for each repository are fully listed in Table 6.

We deliberately limit the search scope for repositories to those predominantly written in Python. Following precedents, focusing on Python repositories allowed us to form assumptions about installation and testing procedures (e.g. repository is organized as a PyPI package, `pytest` is the testing framework) that made scaling up automatic repository setup with SWE-agent more tractable. A worthwhile direction to consider for future work is expanding the coverage of repositories to be more comprehensive of codebases written in different programming languages, as Yang et al. [50] does, extending SWE-bench style evaluation to JavaScript repositories with multimodal inputs.

**Automated repository installation.** The goal of this step is to first, get the installation and testing instructions for a repository, and second, create a Docker image containing the repository with the development environment set up.

Table 6: License associated with each repository as of April 8, 2025. All licenses are permissive and allow for public, nonprofit use.

| | |
|---|---|
| Apache License 2.0 | `Project-MONAI/MONAI; alanjds/drf-nested-routers; arrow-py/arrow; buriy/python-readability; facebookresearch/fvcore; getmoto/moto; google/textfsm; iterative/dvc; jax-ml/jax; jd/tenacity; kayak/pypika; modin-project/modin; pyca/pyopenssl; spulec/freezegun; tkrajina/gpxpy; tornadoweb/tornado; weaveworks/grafanalib` |
| BSD 2-Clause "Simplified" License | `madzak/python-json-logger; pyasn1/pyasn1; pygments/pygments; sunpy/sunpy` |
| BSD 3-Clause "New" or "Revised" License | `Suor/funcy; alecthomas/voluptuous; andialbrecht/sqlparse; cookiecutter/cookiecutter; dask/dask; django/channels; django/daphne; encode/starlette; gawel/pyquery; gweis/isodate; john-kurkowski/tldextract; lepture/mistune; oauthlib/oauthlib; pallets/click; pallets/flask; pallets/jinja; pallets/markupsafe; pandas-dev/pandas; scrapy/scrapy; theskumar/python-dotenv` |
| GNU General Public License v3.0 | `Cog-Creators/Red-DiscordBot; adrienverge/yamllint` |
| GNU Lesser General Public License v2.1 | `chardet/chardet; paramiko/paramiko; pylint-dev/astroid` |
| GNU Lesser General Public License v3.0 | `Knio/dominate` |
| ISC License | `kennethreitz/records` |
| MIT License | `amueller/word_cloud; borntyping/python-colorlog; bottlepy/bottle; cantools/cantools; cdgriffith/Box; cknd/stackprinter; conan-io/conan; cool-RR/PySnooper; datamade/usaddress; dbader/schedule; erikrose/parsimonious; facebookresearch/hydra; facelessuser/soupsieve; getnikola/nikola; graphql-python/graphene; hukkin/tomli; jaraco/inflect; jawah/charset_normalizer; joke2k/faker; keleshev/schema; life4/textdistance; luozhouyang/python-string-similarity; marshmallow-code/apispec; marshmallow-code/marshmallow; marshmallow-code/webargs; martinblech/xmltodict; matthewwithanm/python-markdownify; mewwts/addict; mido/mido; mozillazg/python-pinyin; msiemens/tinydb; pdfminer/pdfminer; pndurette/gTTS; pudo/dataset; pydantic/pydantic; pyparsing/pyparsing; pytest-dev/iniconfig; python-hyper/h11; python-jsonschema/jsonschema; python-openxml/python-docx; pyupio/safety; pyvista/pyvista; r1chardj0n3s/parse; rsalmei/alive-progress; rubik/radon; rustedpy/result; scanny/python-pptx; seatgeek/thefuzz; sloria/environs; sqlfluff/sqlfluff; termcolor/termcolor; tobymao/sqlglot; tox-dev/pipdeptree; tweepy/tweepy; un33k/python-slugify; vi3k6i5/flashtext` |
| Other | `Mimino666/langdetect; PyCQA/flake8; agronholm/exceptiongroup; agronholm/typeguard; aio-libs/async-timeout; benoitc/gunicorn; cloudpipe/cloudpickle; davidhalter/parso; django-money/django-money; gruns/furl; kurtmckee/feedparser; lincolnloop/python-qrcode; mahmoud/boltons; mahmoud/glom; mozilla/bleach; pexpect/ptyprocess; prettytable/prettytable; pwaller/pyfiglet; pydata/patsy; pydicom/pydicom; python-trio/trio; python/mypy; pyutils/line_profiler; seperman/deepdiff` |

We provide the system prompt given to SWE-agent that asks it to install a repository in Figure A.2. Each repository installation task is initialized with a clone of the original repository. No additional steps (e.g. `pypi` package downloads, `conda` environment setup) are performed.

We run SWE-agent with `claude-3-5-sonnet-20241022` with a maximum cost limit of $2 and a maximum call limit of 150. The installation run terminates whenever one of these conditions is met.

For every run, we record the interactions. We then manually review the trajectory, identifying the appropriate installation and testing specifications.

Each run incurs an average cost of $0.72 and an average of 17 steps before SWE-agent issues the `submit` command. The runs typically finish within two minutes. The majority of Python repositories require fewer steps — typically, SWE-agent will view the `CONTRIBUTING.md`, run the installation command provided verbatim in the text, and then runs `pytest`, showing all tests passing. A minority of repositories will require several steps because additional dependencies must be installed with `apt-get`. The manual review process following this requires 3 to 20 minutes. One author carried out this effort for 128 repositories, taking an estimated 18 human hours to accomplish. In the process of reaching 128 repositories, the author gave up on 17 repositories at the manual review stage.

---

**System prompt for generating bugs with an LM**

```
<uploaded_files>
{{working_dir}}
</uploaded_files>
```
I've uploaded a python code repository in the directory `{{working_dir}}`.

Can you please install this repository? Your goal should be to configure the repository's development environment such that existing tests pass. You are currently in the root directory of the repository, and nothing has been installed yet. You in an Ubuntu 22.04 environment.

The repository is predominantly written in Python. Here are several tips for installing it:

1. A good place to start is to look for a `CONTRIBUTING.[md|rst]` file, which will often contain instructions on how to install the repository and any dependencies it may have. Occasionally, the `README.md` file may also contain installation instructions.

2. Usually, a repository may have `setup.py` or `pyproject.toml` files which can be used to install the package. `pip install -e .` is commonly used, although many packages will also require an additional specifier that installs development packages as well (e.g. `pip install -e .[dev]`).

3. To check whether the repository was installed successfully, run tests and see if they pass. You can usually find tests in a `tests/` or `test/` directory. You can run tests using `pytest` or `unittest`, depending on the framework used by the repository.

4. Sometimes, you will need to install additional packages, often listed in a `requirements.txt` or `environment.yml` file. Also, be mindful of Ubuntu system dependencies that may need to be installed via `apt-get` (e.g. `sudo apt-get install <package>`).

Once you are finished with installing the repository, run the `submit` command to submit your changes for review.

---

### A.3 Validation, Evaluation Harnesses

We adapt SWE-bench's validation script to convert each bug patch into a SWE-bench style task instance. This step ensures SWE-smith can be run by existing SWE-bench solutions. The conversion involves two steps. First, the bug patch is applied and pushed as a branch to a mirror clone of the repository. Second, we create a SWE-bench style task instance from the bug patch, populating important fields such as Fail-to-Pass and Pass-to-Pass tests with information from the validation logs.

# B  Bug Generation Strategies

In this section, we review each of the bug generation strategies we employ in depth. While we experimented with several bug generation strategies, the ones we elect to include are those we found to satisfy several desirable properties.

1. The approach works in a codebase-agnostic manner.
2. The approach reliably yields usable task instances (meaning 1+ passing tests break).
3. The approach is controllable; via each strategy's parameters, we can affect the quantity and quality of the generated bugs.

---

### System prompt for generating bugs with an LM

You are a software developer doing chaos monkey testing. Your job is to rewrite a function such that it introduces a logical bug that will break existing unit test(s) in a codebase.
To this end, some kinds of bugs you might introduce include:

(Per inference call, only 3 of the following tips are randomly selected and shown)
- Alter calculation order for incorrect results: Rearrange the sequence of operations in a calculation to subtly change the output (e.g., change (a + b) * c to a + (b * c)).
- Introduce subtle data transformation errors: Modify data processing logic, such as flipping a sign, truncating a value, or applying the wrong transformation function.
- Change variable assignments to alter computation state: Assign a wrong or outdated value to a variable that affects subsequent logic.
- Mishandle edge cases for specific inputs: Change handling logic to ignore or improperly handle boundary cases, like an empty array or a null input.
- Modify logic in conditionals or loops: Adjust conditions or loop boundaries (e.g., replace <= with <) to change the control flow.
- Introduce off-by-one errors in indices or loop boundaries: Shift an index or iteration boundary by one, such as starting a loop at 1 instead of 0.
- Adjust default values or constants to affect behavior: Change a hardcoded value or default parameter that alters how the function behaves under normal use.
- Reorder operations while maintaining syntax: Rearrange steps in a process so the function produces incorrect intermediate results without breaking the code.
- Swallow exceptions or return defaults silently: Introduce logic that catches an error but doesn't log or handle it properly, leading to silent failures.

Tips about the bug-introducing task:
(At inference time, tips are randomly shuffled)
- It should not cause compilation errors.
- It should not be a syntax error.
- It should be subtle and challenging to detect.
- It should not modify the function signature.
- It should not modify the documentation significantly.
- For longer functions, if there is an opportunity to introduce multiple bugs, please do!" - Please DO NOT INCLUDE COMMENTS IN THE CODE indicating the bug location or the bug itself.

Your answer should be formatted as follows:

Explanation: <explanation>

Bugged Code:
```
<bugged_code>
```

---

## B.1  Generating with an LM

We describe our workflows for generating bugs with an LM. For each function or class in a codebase, we prompt an LM to generate either a rewrite that introduces bugs or a complete re-implementation from scratch. This strategy is illustrated in Figure 10.

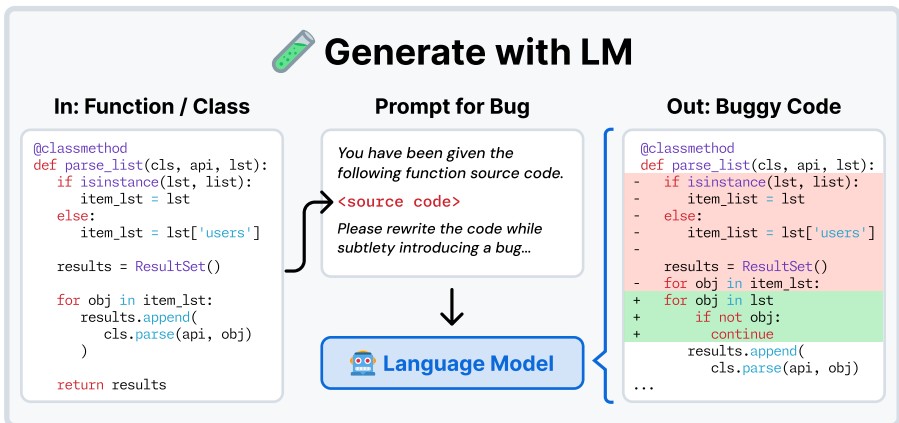

Figure 10: Workflow to generate bugs for a function or class with an LM. We first extract all functions or classes from a codebase, then enumerate across all candidates and prompt the LM to generate either a bug-laced rewrite or a re-implementation.

**Modify existing functions.** Given a Python codebase, we use the `ast` library to identify all unique functions, excluding any functions found under a testing related directory (e.g. `tests`, `testing`). Next, given a function, the LM is asked to write a new version that introduces logical, runtime bugs. Within the prompt, shown in Figure B, several suggestions of types of bugs along with a demonstration of a rewrite are provided.

---

**Prompts for reimplementing bugs with an LM**

**System Prompt**
You are a software developer and you have been asked to implement a function.

You will be given the contents of an entire file, with one or more functions defined in it. Please implement the function(s) that are missing. Do NOT modify the function signature, including the function name, parameters, return types, or docstring if provided. Do NOT change any other code in the file. You should not use any external libraries.

**Task Instance Prompt**
Please implement the function func_signature in the following code:

{file_src_code}

Remember, you should not modify the function signature, including the function name, parameters, return types, or docstring if provided. Do NOT change any other code in the file. Format your output as:

[explanation]

{func_to_write}

---

In our experiments, we use OpenAI's o3 mini model [28] (`o3-mini-2025-01-31`) as the main base model for bug generation. Based on our empirical observations of an LM's tendencies, we include several explicit guidelines in the prompt about what the rewrite should not do. Notably, it is important to ask the LM to not generate any inline comments denoting the location of a bug; we observe that without explicitly specifying this, model generation outputs tend to have inline comments pointing out the bug. We also want to avoid the complexities of identifying and removing such comments from a file diff representation. Second, we state that rewrites causing compilation or syntax errors (e.g. undeclared variables, function definition modifications) should be avoided because such bugs are relatively trivial to solve. We do not experiment extensively with different prompts or generating multiple buggy rewrites per function.

**Modify existing classes.** This method involves a simple amendment to the function rewriting approach. Instead of identifying unique functions (`ast.FunctionDef`), the codebase traversal logic instead looks for classes (`ast.ClassDef`). Otherwise, all other aspects of the implementation are

near identical to function rewriting, with minor changes to the prompt to make bug suggestions and the demonstration more class oriented.

**Rewrite existing functions.** Instead of providing an LM with the original function, we explore an alternative strategy of asking an LM to re-implement a function from scratch. Similar to above, we again use the `ast` library to identify all unique functions. However, instead of directly asking for a bug, we remove the function's implementation, then prompt the LM with the entire file containing the function (minus the original implementation). In the task description, we then explicitly ask for the LM to implement the function without changing the function signature.

## B.2 Procedural Modification

We explore a zero-cost approach to create bugs by performing random modifications to the `ast` representation of a function or class. A "procedural modification" refers to a function that takes in an `ast` and applies a fixed transformation to it, such as removing a loop or swapping the blocks of an if/else clause. This strategy is illustrated in Figure 11.

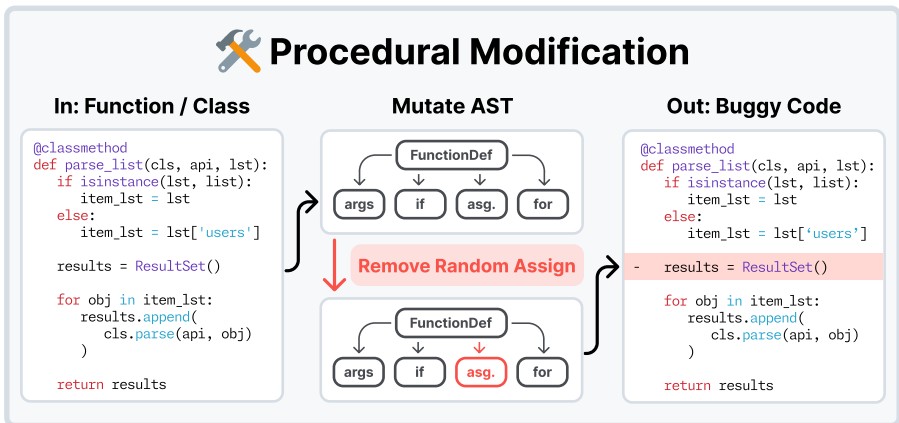

Figure 11: Workflow to generate bugs via procedural modifications. Per function/class, the source code is first convert into an `ast`. The modification then mutates the `ast` (e.g. removes an assignment statement). The `ast` is then converted back into source code with the specific modification introduced.

Similar to the workflow for generating bugs with an LM, we first identify all functions or classes in a repository. Per procedural modification, we first impose a set of criteria that filters out any candidates for which the modification would be impossible. For instance, if the procedural modification removes a random conditional from a function, the modification's criteria will filter out any candidates that are not functions or do not have a conditional. For the remaining candidates, the procedural modification is applied with controlled `likelihood`, where `likelihood` is a fraction indicating how often the procedural modification is applied within a candidate. For example, if the procedural modification removes a random function with a `likelihood` of $0.5$, then for every conditional declared within the function, there is a $50\%$ chance it gets removed. We introduce `likelihood` so procedural modifications do not lead to changes that are too difficult. Finally, the modified `ast` is converted back into source code.

Table 7 is a complete list of filtering criteria that is used for any procedural modification. For the `filter_min_complexity` and `filter_max_complexity` criteria, we define a simple definition of "complexity" as a sum of the number of conditional blocks, loops, boolean operators, exception handling blocks, and comparison operators in a function. The purpose of `filter_min_complexity` is to remove both simple, uninteresting functions (e.g. getter, setter methods) from consideration. `filter_max_complexity` is occasionally used to avoid changing long, monolithic functions.

Table 8 is an exhaustive list of all procedural modifications used to create bugs in a codebase.

Table 7: Pool of criteria used to filter for functions or classes with specific properties. Per procedural modification, a subset of these criteria is first used to filter functions and/or classes from a codebase. The modification is then run on the remainder.

| Index | Criteria | Description |
|---|---|---|
| 1 | `filter_functions` | Is the `ast` a function definition |
| 2 | `filter_classes` | Is the `ast` a class definition |
| 3 | `filter_classes_has_base` | Is the `ast` a class definition with parents |
| 4 | `filter_loops` | Does the `ast` contain a `For` or `While` loop? |
| 5 | `filter_conditionals` | Does the `ast` contain a conditional block? |
| 6 | `filter_assignments` | Is the `ast` a function def. with assignments? |
| 7 | `filter_wrappers` | Does the `ast` contain `try` or `with` blocks? |
| 8 | `filter_if_else` | Does the `ast` contain an `if-else` block? |
| 9 | `filter_operators` | Does the `ast` contain binary, boolean operators? |
| 10 | `filter_min_complexity` | Is the `ast` $\geq$ a complexity score? |
| 11 | `filter_max_complexity` | Is the `ast` $\leq$ a complexity score? |

## B.3 Combine Bug Patches

We discuss the two strategies we use to combine bug patches from the same file or the same module. In practice, we combine LM and procedurally generated bugs that have been validated successfully as usable task instances.

**From the same file.** If two or more functions are defined within a single file, this strategy merges the function-level bug patches together. Given $n$ function-level bugs and $k$ as the number of bugs to combine, there are $\binom{n}{k}$ unique file-level candidate bug patches, which can be a large search space to cover. To make the search space tractable, ensure no single function-level bug is repeatedly used, and generate instances that reliably have 1+ Fail to Pass tests, we implement the following approach described in Algorithm 1.

For each file in a codebase, we first identify the function-level bugs (or bug patches) that edit that file. The pool of bugs we draw from have been *validated*, meaning we have already ensured there is 1+ Fail to Pass test(s) associated with the bug. From these pool of `file_bugs`, the `get_combos` function then generates up to `max_combos` sets of bugs, where the size of each set is `num_bugs`. For each `combo`, or set of bugs, the bugs are applied to the codebase one by one. If all patches are successfully combined, this means they were successfully merged, and the merged patch, which

Table 8: The 13 procedural modification techniques we use to create bugs in a codebase. The "Criteria" column contains indices referencing the corresponding filter defined in Table 7. There are four informal categories — Class, Control Flow, Expressions, Removal — which indicates the general type of modification being made.

| Procedural Modification | | Criteria | Description |
|---|---|---|---|
| Class | Remove Functions | 2, 10 | Removes method(s) + reference(s). |
| | Remove Parent | 3, 10 | Removes base class from class header. |
| | Shuffle Methods | 2, 10 | Shuffles method definitions in a class. |
| Control Flow | Invert If/Else | 8 | Inverts the if-else bodies of a condition. |
| | Shuffle Lines | 11, 12 | Shuffles the lines of a function. |
| Expressions | Change Constants | 1, 9, 10 | $\pm 1$ to a constant numeric value. |
| | Break Chains | 1, 9, 10 | Removes operator(s), operator(s). |
| | Swap Operands | 1, 9, 10 | Mixes order of operands. |
| | Change Operator | 1, 9, 10 | Changes operator(s) (e.g. $+$ to $-$). |
| Removal | Loops | 1, 4, 10 | Remove loops (e.g. `for`, `while`). |
| | Conditionals | 1, 5, 10 | Remove conditionals (`if`). |
| | Assignments | 1, 6, 10 | Remove assignment statements. |
| | Wrappers | 1, 7, 10 | Remove exception (`try`), context (`with`). |

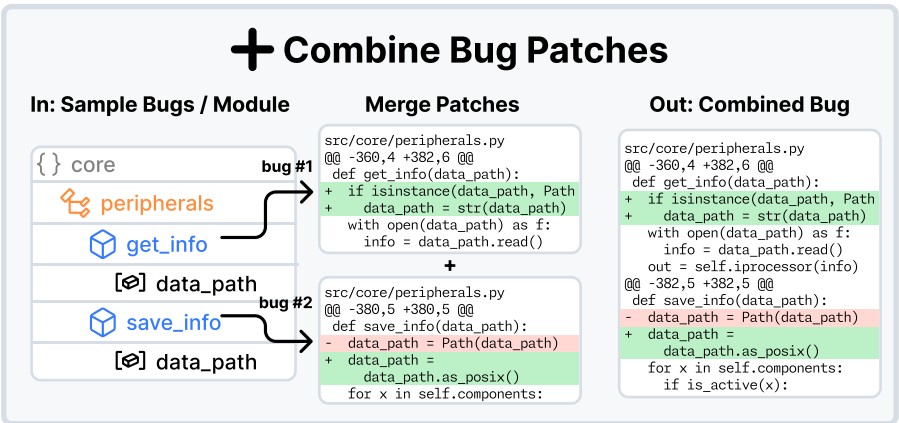

Figure 12: Workflow to generate bugs by combining bug patches. We take $n$ patches (generated using an LM or procedural modification), then sequentially apply each bug patch to the codebase. If all individual patches apply successfully, we save the resulting single patch which now represents all $n$ bugs combined.

---

**Algorithm 1** Combine multiple patches from the same file.

---

**Require:** $codebase, bugs; num\_bugs, limit\_per\_file; max\_combos$
**Ensure:** $min\_bugs \geq 2;$
  $max\_bugs \geq min\_bugs;$
  **procedure** COMBINEFILEBUGS
    **for** each $file$ in $codebase$ **do**
      $file\_bugs \leftarrow$ bugs that apply to $file$
      $combinations \leftarrow$ get_combos($file\_bugs, num\_bugs, max\_combos$)
      **for** each $combo$ in combinations **do**
        Apply $combo$ to $codebase$
        **if** success **then**
          Save $combo$ to disk
          **if** $limit\_per\_file$ reached **then**
            **break**
          **end if**
          $combinations \leftarrow$ [c for c in $combinations$ if c $\cap combo = \emptyset$]
        **end if**
      **end for**
    **end for**
  **end procedure**

---

consists of multiple function-level bugs, is saved and re-validated as a single bug. Merging patches occasionally fails if there is an overlapping conflict between two files, akin to a merge conflict with `git`; this usually happens when a function is declared within another. To ensure a function-level bug is only used once, any remaining bug sets in `combinations` using any patch in `combo` are removed.

The `limit_per_file` and `max_combos` parameters prevent any one file from being over-represented and constrains an otherwise combinatorial large search space. We run this algorithm across all codebase files, typically setting `num_bugs`$= [2, 4]$, `limit_per_file`$= 3$, `max_combos`$= 40$. Decreasing `num_bugs` or increasing the other three parameters improves the yield.

**From the same module.** There are several ways one could imagine composing function-level bugs from multiple bugs, such as combining those that break the same test or have a programmatic relationship (e.g. function a calls function b). We found a relatively straightforward and effective approach to be combining files that edit the same "module". By "module" we are referring to a subdirectory within the source code (e.g. `sklearn/feature_extraction`, `astropy/convolution`). Out of all SWE-bench instances that edit 2+ files, 75% modify files within the same submodule, suggesting a high degree of intra-module code changes. The implementation for our approach is described in Algorithm 2

**Algorithm 2** Combine multiple patches from the same module.

---

**Require:** $bugs$; $num\_bugs$; $limit\_per\_module$; $max\_combos$; $depth$
**Ensure:** $num\_bugs \geq 2$;
  **procedure** COMBINEMODULEBUGS
      $map\_path\_to\_bugs \leftarrow \{\}$
      **for** each $bug$ in $bugs$ **do**
         $path \leftarrow$ get_path_from(bug)
         $map\_path\_to\_patches[path] \leftarrow [bug]$
      **end for**
      Collapse nested paths based on $depth$
      **for all** $(path, patches)$ in $map\_path\_to\_patches$ **do**
         $combinations \leftarrow$ get_combos(patches, $num\_bugs$, $max\_combos$)
         **for** each $combo$ in $combinations$ **do**
            Apply $combo$ to $codebase$
            **if** success and num_files_changed(combo) $\geq 2$ **then**
               Save $combo$ to disk
               **if** $limit\_per\_module$ reached **then**
                  **break**
               **end if**
               $combinations \leftarrow$ [c for c in $combinations$ if c $\cap$ $combo = \emptyset$]
            **end if**
         **end for**
      **end for**
  **end procedure**

---

The implementation for this approach is similar to Algorithm 1 with two key changes. First, we do not do file-by-file or folder-by-folder traversal. Instead, using the diff patches, we create a dictionary `map_path_to_bugs` that mimics the file structure of a codebase. For example, if bug modifies path `a/b/c/d.py`, it is represented as `map_path_to_bugs[a][b][c][d.py]` = `[bug]`. Additional bugs that modify the same path are appended to the list. Since every bug is a function-level bug, there will never be a bug registered in multiple lists. We then "collapse" up to `depth` indices. So for instance, at `depth` $= 3$, the above data structure is collapsed into `map_path_to_bugs[a/b/c][d.py]` = `[bug]`. Finally, any nested dictionaries are collapsed into a single list of patches (e.g. `map_path_to_bugs[a/b/c]` = `[bug]`). Mirroring the procedure in Algorithm 1, we then iterate across this dictionary's values (lists of bugs). Second, we only save patches that modify $2+$ files; aggregate bugs (represented by `combo`) modifying a single file are not considered.

Again, we run this strategy across all 100 repositories, with parameters `num_bugs`$= [2, 5]$, `limit_per_module`$= 10$, `max_combos`$= 100$, and `depth`$= 2$. Reducing `num_bugs`, `depth` and increasing the other parameters yields more bugs. We choose a `depth` of 2 because empirically, we find that meaningful modules are usually declared as immediate sub-folders of the main source code folder (e.g. in `sklearn/feature_extraction`, `sklearn` is the source code folder while `feature_extraction` is the module). A shallower depth leads to less meaningful groupings, while yield decreases significantly for every increased level of depth, particularly for smaller repositories.

## B.4   Pull Request Mirroring

We finally discuss the fourth and last strategy for generating bugs - mirroring real world pull requests (PR). We visualize this process in Figure 13.

**Why use an LM?** When we initially implemented this approach, we attempted to directly perform a `git apply -reverse [patch]` on the codebase. However, for the large majority of patches, this fails. We performed troubleshooting by inspecting 100 PR patches on the `sqlfluff/sqlfluff` repository, leading us to two observations.

1. The majority of these PRs reflect changes that remain present in the codebase today (making the bug creation promising).

2. However, many patches can not be reversed because the exact location (e.g. lines, file) of the relevant code changed because of other changes.

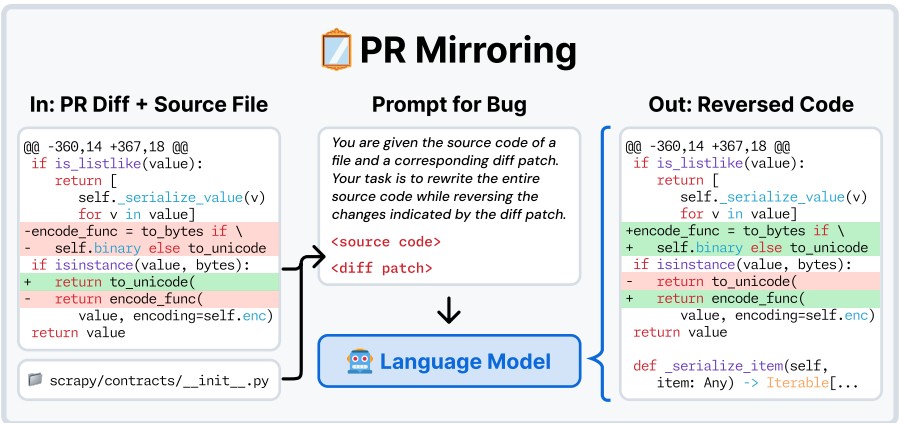

Figure 13: Workflow to generate bugs by reverting changes made in the diff patch corresponding to a real GitHub pull request (PR). Given the patch and the files modified by the patch, we prompt the LM to generate a complete rewrite of each file that *reverses* the changes made in the PR. The changes are applied to the codebase, and we extract the patch, which now captures the reversal of the PR changes.

Therefore, we employ LMs to perform patch reversal, and find that reasoning models (e.g. `o3-mini` [28]) are particularly effective.

**Description of method.** We follow SWE-bench's methodology for crawling PRs created January 1st, 2023 and onwards, with minor and arbitrary exceptions for some repositories where we crawl older PRs as well. Per PR, we iterate across the file(s) changed by the patch. Per file, we prompt an LM with the file-specific changes from the patch along with the file's source code in the current state of the repository (*not* the repository's state corresponding to when the PR was applied, referred to as the `base_commit` in SWE-bench). The LM is asked to generate a rewrite of the file that reverts the changes reflected in the PR. We aggregate the changes across all file(s) into a single patch.

Because we are interested in problems that our expert trajectory generation method (SWE-agent + Claude 3.7 Sonnet) has a chance of solving, we do not attempt to reproduce PRs that change more than 8 files. This constraint is imposed because no SWE-bench instance that edits more than 6 files has ever been solved [17].

**How well does PR mirroring work?** We scrape the PRs corresponding to 100 randomly selected SWE-bench task instances from the `django/django` GitHub repository and attempt to recreate these task instances with SWE-smith's collection process. We successfully recovered 92 of 100 task instances. Of these, 84 break identical F2P test(s), with the remaining 8 breaking a subset because some tests were removed over time. This sanity check gives us confidence that the PR mirroring strategy lives up to its name.

**Comparison to SWE-bench.** This approach has several benefits and drawbacks compared to SWE-bench's collection pipeline. First, it removes the need to create instance-specific Docker images — all PRs are mirrored against the same version of a repository. This also implies that there is no need to write installation specifications for past versions of a repository, which is typically the most laborious step in task construction with SWE-bench. Finally, this strategy also allows us to loosen the requirements on what PRs we attempt to convert into a task instance. In SWE-bench, the core requirements for what PRs to attempt to convert into a task instance include:

1. It must edit 1+ code files (e.g. not just `.md`, `.rst` files).
2. It must reference 1+ GitHub issues, which serves as the problem statement.
3. It must edit 1+ testing related files (1+ files with a `test`-adjacent keyword in it).

With this collection strategy and SWE-smith's focus on training data, the second and third requirements are no longer necessary. If there is no associated issue, issue text can simply be generated. If the patch does not contain any testing related changes, this is tolerable, as the validation stage will determine whether the PR breaks any tests. With these considerations, we purport that SWE-smith's PR mirroring strategy can re-purpose a higher percentage of real world code changes for training purposes.

The main downside is that the rest of the repository is out of sync with the state of the codebase when the PR was applied. As a result, it's possible that changes in the behavior of the rest of the codebase may affect the issue's reproducibility or the accuracy of the issue description (e.g. line numbers referenced in the issue text are likely somewhat off with respect to the codebase). However, a simple mitigation for this is to create a Docker image for a repository at an earlier commit that's closer to the original creation date of the issue. While we do not carry out a targeted experiment, we hypothesize that using SWE-smith, we would be able to reproduce SWE-bench entirely with 10x less human hours with an estimated 2294 x $0.055 = $126.17 in costs.

## C   Issue Generation

We cover the four issue generation strategies we experiment with to determine issue text's effect on how solvable a SWE-smith instance is along with the trajectory's value as a training data point.

**Generated with LM.** We prompt an LM with a randomly selected SWE-bench Verified problem statement, the bug patch, list of Fail-to-Pass tests, source code for one Fail-to-Pass test, and the execution logs of running all the Fail-to-Pass tests. We ask the LM to generate an issue that describes the bug conveyed in the patch in the style of the SWE-bench Verified demonstration. Figure C shows the system prompt for this strategy.

---

**System prompt for generating issues with an LM**

You are a software engineer helping to create a realistic dataset of synthetic GitHub issues.

You will be given the following input:

1. Demonstration: A realistic GitHub issue to mimic (included in the <demonstration> tag).
2. Patch: A git diff output/PR changes that introduces a bug (included in the <patch> tag).
3. Test output: The output of running the tests after the patch is applied (included in the <test_output> tag).
4. Test source code: Source code for one or more tests that failed (included in the <test_source_code> tag).

Output: A realistic GitHub issue for the patch.

Guidelines:
- Mimic the style and structure of the demonstration issues. If the demonstration issues are not well structured, your output should also be not well structured. If the demonstrations use improper or no markdown, your output should also use improper or no markdown. If the demonstrations are short/long, your output should also be short/long (if possible). If the demonstrations include human "flavor text" or "fluff", your output should also include human "flavor text" or "fluff". Do this even if it conflicts with your default behavior of trying to be extremely concise and helpful.
- DO NOT explain the fix/what caused the bug itself, focus on how to reproduce the issue it introduces
- Do not mention pytest or what exact test failed. Instead, generate a realistic issue.
- If possible, include information about how to reproduce the issue. An ideal reproduction script should raise an error
or print an unexpected output together with the expected output.
However, still include this information in a style very similar to the demonstration issues.

---

**Fixed issue templates.** We create a set of 7 pre-defined issue templates, listed in Table 9. Each template uses information from the bug patch or Fail-to-Pass tests associated with every task instance. Given a dataset of task instances, we randomly select one of the templates to use as the problem statement according to the probabilities listed in Table 9. The reason we assign the highest likelihood for the prompt that provides all four categories of information (bug type, files changed, functions changed, Fail-to-Pass tests) is to ensure that a higher proportion of task instances are well-specified.

**Fail-to-Pass test code and execution logs.** Another approach is showing the source code and test execution logs for a randomly selected Fail-to-Pass test. This approach is motivated by the lack of reproduction code or expected/actual behavior of code communicated with fixed issue templates. We show code and execution logs only for a single Fail-to-Pass test; if a task instance has more than one Fail-to-Pass test, we do not disclose remaining tests.

Table 9: List of issue text templates we use to generate problem statements. Across all templates, four types of information are included — the files with bugs, functions with bugs, Fail-to-Pass test(s), and the type of bug. Templates that offer less information are generally assigned a lower probability.

| Template | Prob. | Information Provided |
|---|---|---|
| Basic | 0.05 | None |
| Files | 0.1 | States which file(s) have bug(s). |
| Funcs | 0.15 | States which file(s) and func(s) have bug(s). |
| Tests | 0.1 | States that some tests are failing. |
| F2P Tests | 0.1 | States which tests are failing. |
| Bug Type | 0.05 | States failure type. |
| Bug Type + Files | 0.15 | States failure type and which file(s) have bug(s) |
| Bug Type + Files + Test | 0.15 | States failure type, which file(s) have bug(s), and a random F2P test. |
| Bug Type + Files + Funcs + Test | 0.15 | States failure type, which file(s) and func(s) have bug(s), and a random F2P test. |

**Original issue text.** This strategy works exclusively for some task instances generated using PR Mirroring. If a PR is successfully mirrored, we use the text from the associated issues as the problem statement, exactly as done in SWE-bench. Of the 2345 task instances represented in SWE-smith mirrored from real-world PRs, 708 or 30.19% of these have one or more associated GitHub issue(s) to create a SWE-bench style problem statement.

# D   Dataset Statistics

We present additional breakdowns and analyses of the SWE-smith dataset, focusing on the kinds of repositories and bugs that are represented.

**Repository categorization.** We present an exhaustive list of repositories used in SWE-smith in Table D. We categorize the repositories into seven general buckets: Data Parsing and Transformation (39), Web & API Development (11), Code Quality & Testing (12), Visualization & Presentation (8), System Tools & Protocols (17), Natural Language Processing (7), and Miscellaneous (6). The categorizations were performed by first, determining an appropriate set of categories based on manual inspection supported by the descriptions and GitHub topics associated with each repository. After settling upon the buckets, we asked GPT-4o to provide a label based on the repository's metadata and `README` dump. SWE-smith represents a wider and more variegated coverage of software tools and applications compared to any prior works.

| Repository | | Description |
|---|---|---|
| *Code Quality and Testing* | | |
| PyCQA/flake8 | | flake8 is a python tool that glues together pycodestyle, pyflakes, mccabe, and third-party plugins to check the style and quality of some python code. |
| Suor/funcy | | A fancy and practical functional tools |
| adrienverge/yamllint | | A linter for YAML files. |
| agronholm/typeguard | | Run-time type checker for Python |
| cknd/stackprinter | | Debugging-friendly exceptions for Python |
| cool-RR/PySnooper | | Never use print for debugging again |
| getmoto/moto | | A library that allows you to easily mock out tests based on AWS infrastructure. |
| pylint-dev/astroid | | A common base representation of python source code for pylint and other projects |
| pytest-dev/iniconfig | None | None |
| pytest-dev/iniconfig | | |
| python/mypy | | Optional static typing for Python |

*Continued on next page*

| Repository | Description |
| --- | --- |
| pyupio/safety | Safety checks Python dependencies for known security vulnerabilities and suggests the proper remediations for vulnerabilities detected. |
| pyutils/line_profiler | Line-by-line profiling for Python |
| rubik/radon | Various code metrics for Python code |
| spulec/freezegun | Let your Python tests travel through time |
| sqlfluff/sqlfluff | A modular SQL linter and auto-formatter with support for multiple dialects and templated code. |

Data Parsing and Transformation

| Repository | Description |
| --- | --- |
| alecthomas/voluptuous | CONTRIBUTIONS ONLY: Voluptuous, despite the name, is a Python data validation library. |
| andialbrecht/sqlparse | A non-validating SQL parser module for Python |
| buriy/python-readability | fast python port of arc90's readability tool, updated to match latest readability.js! |
| burnash/gspread | Google Sheets Python API |
| chardet/chardet | Python character encoding detector |
| cloudpipe/cloudpickle | Extended pickling support for Python objects |
| dask/dask | Parallel computing with task scheduling |
| datamade/usaddress | :us: a python library for parsing unstructured United States address strings into address components |
| davidhalter/parso | A Python Parser |
| erikrose/parsimonious | The fastest pure-Python PEG parser I can muster |
| facelessuser/soupsieve | A modern CSS selector implementation for BeautifulSoup |
| gawel/pyquery | A jquery-like library for python |
| google/textfsm | Python module for parsing semi-structured text into python tables. |
| gruns/furl | URL parsing and manipulation made easy. |
| gweis/isodate | ISO 8601 date/time parser |
| hukkin/tomli | A lil' TOML parser |
| jawah/charset_normalizer | Truly universal encoding detector in pure Python |
| john-kurkowski/tldextract | Accurately separates a URL's subdomain, domain, and public suffix, using the Public Suffix List (PSL). |
| joke2k/faker | Faker is a Python package that generates fake data for you. |
| jsvine/pdfplumber | Plumb a PDF for detailed information about each char, rectangle, line, et cetera — and easily extract text and tables. |
| kayak/pypika | PyPika is a python SQL query builder that exposes the full richness of the SQL language using a syntax that reflects the resulting query. PyPika excels at all sorts of SQL queries but is especially useful for data analysis. |
| keleshev/schema | Schema validation just got Pythonic |
| kennethreitz/records | SQL for Humans™ |
| kurtmckee/feedparser | Parse feeds in Python |
| lepture/mistune | A fast yet powerful Python Markdown parser with renderers and plugins. |
| madzak/python-json-logger | Json Formatter for the standard python logger |
| mahmoud/glom | Python's nested data operator (and CLI), for all your declarative restructuring needs. Got data? Glom it! |
| marshmallow-code/marshmallow | A lightweight library for converting complex objects to and from simple Python datatypes. |
| martinblech/xmltodict | Python module that makes working with XML feel like you are working with JSON |
| matthewwithanm/python-markdownify | Convert HTML to Markdown |
| mewwts/addict | The Python Dict that's better than heroin. |
| mido/mido | MIDI Objects for Python |

Continued on next page

| Repository | Description |
| --- | --- |
| modin-project/modin | Modin: Scale your Pandas workflows by changing a single line of code |
| mozilla/bleach | Bleach is an allowed-list-based HTML sanitizing library that escapes or strips markup and attributes |
| msiemens/tinydb | TinyDB is a lightweight document oriented database optimized for your happiness :) |
| pandas-dev/pandas | Flexible and powerful data analysis / manipulation library for Python, providing labeled data structures similar to R data.frame objects, statistical functions, and much more |
| pdfminer/pdfminer.six | Community maintained fork of pdfminer - we fathom PDF |
| pudo/dataset | Easy-to-use data handling for SQL data stores with support for implicit table creation, bulk loading, and transactions. |
| pydantic/pydantic | Data validation using Python type hints |
| pydata/patsy | Describing statistical models in Python using symbolic formulas |
| pydicom/pydicom | Read, modify and write DICOM files with python code |
| pygments/pygments | Pygments is a generic syntax highlighter written in Python |
| pyparsing/pyparsing | Python library for creating PEG parsers |
| python-jsonschema/jsonschema | An implementation of the JSON Schema specification for Python |
| python-openxml/python-docx | Create and modify Word documents with Python |
| r1chardj0n3s/parse | Parse strings using a specification based on the Python format() syntax. |
| scanny/python-pptx | Create Open XML PowerPoint documents in Python |
| scrapy/scrapy | Scrapy, a fast high-level web crawling & scraping framework for Python. |
| seperman/deepdiff | DeepDiff: Deep Difference and search of any Python object/data. DeepHash: Hash of any object based on its contents. Delta: Use deltas to reconstruct objects by adding deltas together. |
| sloria/environs | simplified environment variable parsing |
| sunpy/sunpy | SunPy - Python for Solar Physics |
| tkrajina/gpxpy | gpx-py is a python GPX parser. GPX (GPS eXchange Format) is an XML based file format for GPS tracks. |
| tobymao/sqlglot | Python SQL Parser and Transpiler |
| un33k/python-slugify | Returns unicode slugs |

*Machine Learning and AI*

| Repository | Description |
| --- | --- |
| facebookresearch/fvcore | Collection of common code that's shared among different research projects in FAIR computer vision team. |
| facebookresearch/hydra | Hydra is a framework for elegantly configuring complex applications |
| HIPS/autograd | Efficiently computes derivatives of NumPy code. |
| iterative/dvc | Data Versioning and ML Experiments |
| jaraco/inflect | Correctly generate plurals, ordinals, indefinite articles; convert numbers to words |
| life4/textdistance | Compute distance between sequences. 30+ algorithms, pure python implementation, common interface, optional external libs usage. |
| luozhouyang/python-string-similarity | A library implementing different string similarity and distance measures using Python. |
| Mimino666/langdetect | Port of Google's language-detection library to Python. |
| mozillazg/python-pinyin | 汉字转拼音(pypinyin) |
| pndurette/gTTS | Python library and CLI tool to interface with Google Translate's text-to-speech API |
| Project-MONAI/MONAI | AI Toolkit for Healthcare Imaging |
| seatgeek/thefuzz | Fuzzy String Matching in Python |

*Continued on next page*

| Repository | Description |
| --- | --- |
| vi3k6i5/flashtext | Extract Keywords from sentence or Replace keywords in sentences. |

| *System Tools and Protocols* | |
| --- | --- |
| agronholm/exceptiongroup | Backport of PEP 654 (exception groups) |
| aio-libs/async-timeout | asyncio-compatible timeout class |
| arrow-py/arrow | Better dates & times for Python |
| borntyping/python-colorlog | A colored formatter for the python logging module |
| cantools/cantools | CAN bus tools. |
| conan-io/conan | Conan - The open-source C and C++ package manager |
| cookiecutter/cookiecutter | A cross-platform command-line utility that creates projects from cookiecutters (project templates), e.g. Python package projects, C projects. |
| dbader/schedule | Python job scheduling for humans. |
| gruns/icecream | Never use print() to debug again. |
| jd/tenacity | Retrying library for Python |
| mahmoud/boltons | Like builtins, but boltons. 250+ constructs, recipes, and snippets which extend (and rely on nothing but) the Python standard library. Nothing like Michael Bolton. |
| oauthlib/oauthlib | A generic, spec-compliant, thorough implementation of the OAuth request-signing logic |
| pallets/click | Python composable command line interface toolkit |
| paramiko/paramiko | The leading native Python SSHv2 protocol library. |
| pexpect/ptyprocess | Run a subprocess in a pseudo terminal |
| pyasn1/pyasn1 | Generic ASN.1 library for Python |
| pyca/pyopenssl | A Python wrapper around the OpenSSL library |
| python-hyper/h11 | A pure-Python, bring-your-own-I/O implementation of HTTP/1.1 |
| python-trio/trio | Trio – a friendly Python library for async concurrency and I/O |
| rustedpy/result | NOT MAINTAINED - A simple Rust like Result type for Python 3. Fully type annotated. |
| termcolor/termcolor | ANSI color formatting for output in terminal |
| theskumar/python-dotenv | Reads key-value pairs from a .env file and can set them as environment variables. It helps in developing applications following the 12-factor principles. |
| tox-dev/pipdeptree | A command line utility to display dependency tree of the installed Python packages |

| *Visualization and Presentation* | |
| --- | --- |
| amueller/word_cloud | A little word cloud generator in Python |
| lincolnloop/python-qrcode | Python QR Code image generator |
| prettytable/prettytable | Display tabular data in a visually appealing ASCII table format |
| pwaller/pyfiglet | An implementation of figlet written in Python |
| rsalmei/alive-progress | A new kind of Progress Bar, with real-time throughput, ETA, and very cool animations! |
| weaveworks/grafanalib | Python library for building Grafana dashboards |

| *Web and API Development* | |
| --- | --- |
| Cog-Creators/Red-DiscordBot | A multi-function Discord bot |
| Knio/dominate | Dominate is a Python library for creating and manipulating HTML documents using an elegant DOM API. It allows you to write HTML pages in pure Python very concisely, which eliminate the need to learn another template language, and to take advantage of the more powerful features of Python. |
| alanjds/drf-nested-routers | Nested Routers for Django Rest Framework |

| Repository | Description |
|---|---|
| benoitc/gunicorn | gunicorn 'Green Unicorn' is a WSGI HTTP Server for UNIX, fast clients and sleepy applications. |
| bottlepy/bottle | bottle.py is a fast and simple micro-framework for python web-applications. |
| django-money/django-money | Money fields for Django forms and models. |
| django/channels | Developer-friendly asynchrony for Django |
| django/daphne | Django Channels HTTP/WebSocket server |
| encode/starlette | The little ASGI framework that shines. |
| getnikola/nikola | A static website and blog generator |
| graphql-python/graphene | GraphQL framework for Python |
| marshmallow-code/apispec | A pluggable API specification generator. Currently supports the OpenAPI Specification (f.k.a. the Swagger specification).. |
| marshmallow-code/webargs | A friendly library for parsing HTTP request arguments, with built-in support for popular web frameworks, including Flask, Django, Bottle, Tornado, Pyramid, webapp2, Falcon, and aiohttp. |
| pallets/jinja | A very fast and expressive template engine. |
| pallets/markupsafe | Safely add untrusted strings to HTML/XML markup. |
| tornadoweb/tornado | Tornado is a Python web framework and asynchronous networking library, originally developed at FriendFeed. |
| tweepy/tweepy | Twitter for Python! |

### D.1  Bug Generation Statistics

We provide extensive details about different aspects of each of the bug generation strategies, including the yield rates, labor/monetary costs, and dataset characterizations.

**Yield rates.** In Table 11, we provide the yield rates for each bug generation method across all repositories in SWE-smith. In general, we find that the PR Mirroring has the lowest yield rate at 13.18% (although this rate is somewhat higher than SWE-bench's yield rate of $2294/93139 = 2.46\%$). For using LMs to generate bugs, modifying functions to introduce bugs intentionally has a higher yield than asking LMs to perform a best-effort rewrite. The efficacy of Procedural Modifications varies by strategy. For instance, shuffling the functions declared in a class only breaks existing test(s) 1.93% of the time, but inverting a conditional will lead to a task instance for 47.04% of modifications. Finally, combining bug patches has an extremely high yield rate - this is to be expected because we only attempt to combine bug patches that have been validated as usable task instances breaking 1+ tests.

The number of repositories captured by each bug generation technique varies due to each strategy's specific preconditions, which at times may not be effective for some repositories. For instance, the *Procedural (Class *)* set of methods only mutates Python classes. This strategy is fruitless for the minority of SWE-smith repositories that do not define any classes. The *Procedural (Op Break Chains)* method randomly removes operations and operands from expressions with two or more operations (e.g. $a + b + c \rightarrow a + b$) — such expressions are not always present in SWE-smith repositories.

The collective yield rate across SWE-smith's bug generation strategies is significantly higher than SWE-bench's collection strategy.

The yield rate also varies with respect to the repository it is being applied to. We provide a summary of yield rates by repository in Table 12. We generally observe that lower test coverage correlates with a lower yield rate.

**Dataset characterizations.** In Table 13, we provide statistics about the validated task instances produced by different bug generation strategies. Our work's LM-based strategies rewrite one function in one file. Procedural modifications will also only change one file, but depending on the strategy, 1+ functions or classes may be changed. Combining multiple patches from the same file always produces a patch with 2+ functions edited. Combining across modules produces a patch with 2+ files edited. The targeted nature of each of the bug creation strategies is reflected in the typical number of functions and files that the bugs produced by each strategy edits.

Table 11: Yield rates for different bug generation strategies covered in Section B. We show the number of repositories that each strategy was run on, the number of bug candidates generated by each strategy, and the number of instances, or the number of candidates that were validated to have 1+ Fail to Pass test. The yield rate for

| Strategy | # Repos | # Candidates | # Instances | Yield Rate |
|---|---|---|---|---|
| Combine (file) | 124 | 6020 | 5865 | 97.43% |
| Combine (module) | 65 | 4396 | 4227 | 96.16% |
| LM (Modify) | 108 | 31950 | 17887 | 55.98% |
| LM (Rewrite) | 128 | 11908 | 4173 | 35.04% |
| PR Mirroring | 108 | 6934 | 2344 | 33.8% |
| Procedural (Class Rm Base) | 103 | 1401 | 463 | 33.05% |
| Procedural (Class Rm Funcs) | 103 | 2506 | 1180 | 47.09% |
| Procedural (Class Shuffle Funcs) | 103 | 2504 | 47 | 1.88% |
| Procedural (Ctrl Invert If) | 105 | 4695 | 2321 | 49.44% |
| Procedural (Ctrl Shuffle) | 104 | 9055 | 4015 | 44.34% |
| Procedural (Op Break Chains) | 71 | 747 | 225 | 30.12% |
| Procedural (Op Change Const) | 77 | 723 | 257 | 35.55% |
| Procedural (Op Change) | 81 | 1507 | 450 | 29.86% |
| Procedural (Op Swap) | 87 | 2141 | 483 | 22.56% |
| Procedural (Remove Assign) | 121 | 5470 | 2661 | 48.65% |
| Procedural (Remove Cond) | 120 | 5288 | 2311 | 43.7% |
| Procedural (Remove Loop) | 110 | 1945 | 860 | 44.22% |
| Procedural (Remove Wrapper) | 80 | 884 | 368 | 41.63% |
| All | 129 | 100074 | 50137 | 50.1% |

Table 12: Yield rates for different repositories represented in SWE-smith.

| Yield Rate | # of Repositories |
|---|---|
| 0-25% | 10 |
| 25-50% | 31 |
| 50-75% | 60 |
| 75-100% | 27 |

In Figure 14, we show the distributions for different attributes of SWE-smith compared to other SWE-bench style datasets. Compared to prior works, there is a much higher proportion of task instances with more than one Fail-to-Pass test. For any one repository, we find that SWE-smith task instances collectively cause failures for a much higher percentage of the testing suit than other datasets; a potential benefit of this is that training on SWE-smith based trajectories may expose models to a much broader set of functionalities in a codebase. The number of lines and files edited by SWE-smith task instances is highly similar to the trend lines for SWE-bench Verified.

We note that unlike other datasets, the trend line of SWE-smith task instances is "adjustable". In other words, the Figure 14 distributions are a capture of the task instances provided in this release of SWE-smith. However, because of SWE-smith's flexible bug creation techniques, the distribution can be "shaped" if needed. For instance, generating more task instances using the bug patch combination method would shift all three curves in Figure 14. We make this point to highlight the fact that the attributes of SWE-bench task instances are, in a sense, constrained by real world software development behavior. On the other hand, SWE-smith can be used to break tests and code that may not be reflected at all in any existing pull request. In this sense, we argue that LMs trained on SWE-smith have better "exposure" to a codebase compared to exclusively training on pull requests.

**Continuation of scaling execution environments.** The validation and evaluation procedures for SWE-smith deviate slightly from SWE-bench's harnesses. The main reasons for these differences can largely be attributed to the granularity of installation specifications. In SWE-bench, each task instance corresponds to a unique base commit, with additional `version` and `environment_setup_commit` keys needed as indirection for mapping an instance to the correct set of installation and testing

Table 13: Statistics for attributes of a SWE-smith task instance across different bug generation strategies, reported as *median (IQR)*, where IQR is the inter-quartile range (25th–75th percentile).

| Strategy | # Instances | # F2P | $\Delta$ Lines | $\Delta$ Functions | $\Delta$ Files |
|---|---|---|---|---|---|
| Combine | 10092 | 15 (5-48) | 19 (12-36) | 2 (2-3) | 1 (1-2) |
| LM | 22060 | 4 (1-17) | 6 (3-15) | 1 (1-1) | 1 (1-1) |
| PR Mirroring | 2344 | 3 (1-14) | 20 (8-55) | 2 (2-4) | 1 (1-2) |
| Procedural | 15641 | 7 (2-32) | 7 (5-15) | 1 (1-1) | 1 (1-1) |

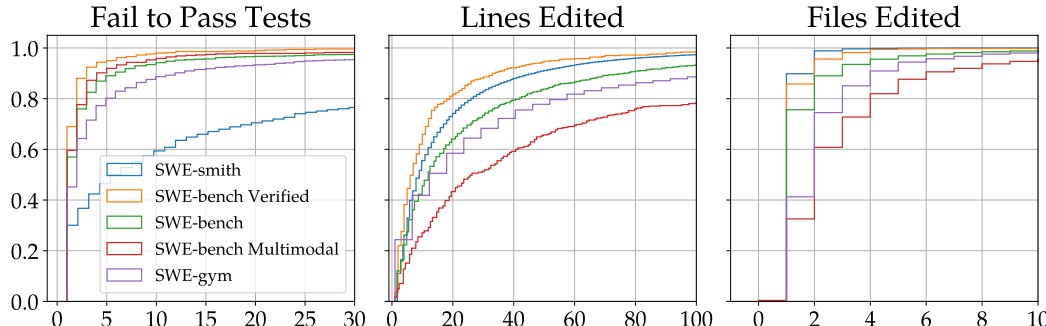

Figure 14: Comparison of cumulative distributions for Fail-to-Pass tests along with the lines and files edited by the gold patch across SWE-smith and four SWE-bench style datasets.

instructions. Across time, the continuous evolution of a repository and its dependencies make for an incredibly high degree of variability in how a repository should be installed correctly. To solve this variability, the community has resorted to creating an image per task instance, as done in Chowdhury et al. [9]. Therefore, for 2294 SWE-bench task instances, there are 2294 unique Docker images, each at a size of at least several gigabytes ($\sim$ 5-6 GBs).

On the other hand, the simplicity and scalability of SWE-smith's design allows one to support many task instances with comparatively much fewer Docker images. As mentioned above, installation and testing procedures are (repository, commit) specific. Therefore, when bugs are generated from each (repository, commit), all bugs can be reproduced and tested successfully from the same Docker image. In other words, if I generate 100 bugs for a repository at some commit, instead of 100 Docker images, only a single Docker image is required to run inference on any of the 100 task instances.

This design is what enables SWE-smith to be significantly more space-efficient than SWE-bench. Based on the publicly released images, for SWE-bench's 2294 task instances, 1.2 TBs of storage are required to download all Docker images locally. for SWE-bench Multimodal's 517 task instances, 1.2 TBs are required. The higher per-instance Docker image size for SWE-bench Multimodal is due to how JavaScript dependency management tools (e.g. `npm`) require more storage compared to equivalent Python infrastructure (e.g. `pypi`). Pan et al. [31] states that each image for the 2438 instances an average of 2.6GB, totaling 6 TB of storage total. Such a storage requirement can be a significant barrier for academic practitioners.

On the other hand, with more than 20x the number of bugs, SWE-smith requires only 125 Docker images total, corresponding to the number of unique (repository, commit) pairs (in this work, for each repository, we only determine installation and test specifications for one commit). The 125 images require a total of $290.54$ GBs. In summary, compared to SWE-bench's task collection strategy, SWE-smith's design makes it easier to not only create task instances, but also train on them as well.

### D.2 Case Study: SWE-bench & SWE-smith

To better understand the differences between the SWE-bench and SWE-smith collection strategies, we perform SWE-smith collection on the `pallets/flask` GitHub repository, one of the 12 test split repositories from the original SWE-bench benchmark. We review the steps covered in Section 2.1 applied to `pallets/flask` in detail. First, we defined the installation and testing specifications

for the `pallets/flask` repository at commit `bc09840`. Next, we apply the LM modification bug generation strategy to this version of the repository, generating 267 unique bugs.

We observe several differences. First, *the SWE-smith collection strategy yields a much higher number of bugs outright.* From SWE-bench, 11 task instances are from the `pallets/flask` repository. The task instances were originally filtered from 2434 pull requests (PRs), with 107 satisfying SWE-bench's filtering criteria of (1) being linked to one or more issues and (2) featuring 1+ new tests. Out of these 107, the 11 (0.45% of 2434) task instances represent the proportion of PRs that execution environments could be successfully constructed for. On the other hand, running the function-level rewriting strategy for bug generation originally yielded 402 candidates, of which 267 were determined to be valid task instances.

Second, *SWE-smith requires significantly less human effort while only incurring minor costs.* Collecting the 11 `pallets/flask` task instances (steps include scraping PRs, determining repository versions across time, defining version-specific installation/test specifications, running execution-based validation multiple times) took an estimated 38 hours worth of human labor. On the contrary, defining installation and testing specifications for the latest commit of `pallets/flasks` took 10 minutes. The subsequent function-level rewriting strategy for bugs took 23 minutes to run, incurring a total cost of just $2.47 ($\sim$\$0.00613 per instance). The final execution-based validation step that filters out $402 - 267 = 135$ unqualified bug candidates ran in 14 minutes. Since both the bug and problem statement generation strategies are repository agnostic, no additional human intervention is necessary for these steps. Head to head, per instance for the `pallets/flask` repository, SWE-bench style collection requires $38 \times 60/11 = 207.27$ minutes compared to 0.176 minutes ($\sim 10.6$ seconds) and $0.00613 in API costs using SWE-smith.

Third, *collectively, SWE-smith task instances break a significantly larger proportion of existing tests in a codebase.* We define "bug coverage" as the proportion of tests broken by 1+ instance across all task instances. For the SWE-bench split of `pallets/flask`, there are 207 unique tests across all 11 instances. Of these 207 tests, 15 are broken by 1+ instance, corresponding to a bug coverage rate of 7.25%. For the SWE-smith split of `pallets/flask`, there are 474 unique tests across 267 instances. The larger amount of tests is due to increased test coverage in the `pallets/flask` repository as of Nov. 28, 2024 (when SWE-smith was collected) compared to June 2023 (when SWE-bench was collected). Of these 474 tests, 422 are broken by 1+ instance, a bug coverage rate of 89.03%. We attribute the significant difference to a consistent tendency in real world open source software development workflows, that is, the *minority* of tests are introduced to capture existing, errant behavior in the repository. The significant majority of tests are committed alongside working code, ensuring that already correct behavior is upheld. Well-maintained repositories will typically not merge commits that cause such tests to fail. This results in a large number of tests where few to no commits correspond to those tests' failures.

Finally, *SWE-smith does not yield instances appropriate for evaluation.* The SWE-smith pipeline as presented does not produce hidden tests, a crucial difference that makes SWE-bench more suitable for evaluation. Consequently, when expert trajectories are generated, the Fail-to-Pass tests are present in the repository at inference time. Furthermore, our issue generation strategy does not include checks for known problems such as underspecified text descriptions or solution leakage [9]. Simple amendments could make SWE-smith task instances suitable for evaluation, such as deleting Fail-to-Pass test functions or files along with a validation procedure around the ambiguity and leakage of the issue text. Finally, thorough analyses of how faithful SWE-smith task instances are to real world issues and PRs would be necessary to justify synthetic bugs for evaluation.

# E   Difficulty Rating

We train a model that labels a task with one of three difficulty labels: $< 15$ minutes (easy), 15 minutes - 1 hour (medium), and 1+ hour (hard). This model allows us to quantify the difficulty of individual task instances and, in aggregate, the difficulty of entire datasets.

To train this model, we use 1699 annotations from Chowdhury et al. [9]. In their work towards curating SWE-bench Verified, a subset of 1699 SWE-bench task instances were labeled with four difficulty levels: $< 15$ min, 15 min - 1 hr, 1-4 hrs, and 4+ hrs. Generally, three annotators were assigned to each instance, and the difficulty annotations were ensembled by taking the majority

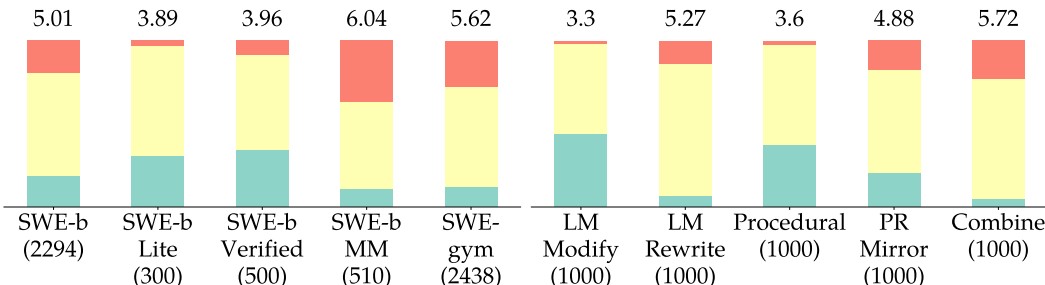

Figure 15: Distribution of task instance difficulty (easy/medium/hard) for existing SWE-bench style datasets (left 5 bars) and SWE-smith (right 5 bars), assessed by our difficulty rating model. The average difficulty score for each dataset is listed above each bar. For SWE-smith, per bug strategy, we sample 1000 task instances with LM generated issue text.

choice for a sample, or the median if there is no majority. The distribution of annotated difficulties, from easiest to hardest, is 24.5%, 53.5%, 19.4%, and 2.8%.

Because there are very few samples in the 4+ hr category, we reclassify the 1-4 hr and 4+ hr instances into a single 1+ hr category. Next, we create corresponding train and test datasets at a 80/20% split, randomly shuffling the instances while ensuring the train and test distributions do not deviate significantly from the original. An instance's problem statement and solution patch are provided as input, and one of the three difficulty labels serves as the target output. We perform LoRA fine-tuning [13] on a Qwen 2.5 32B Instruct model using the Unsloth [10] library. The model achieves an accuracy of 75.3% on the test set. All errant predictions are off by one; in other words, the model never predicted $< 15$ min when the label was 1+ hr, and vise versa.

Using this model, we can grade the difficulty of a SWE-smith instance once the bug patch and corresponding issue text have been created. To provide a succinct summary of difficulty for a dataset of SWE-bench style task instances, we propose a "difficulty score" metric. Each label corresponds to a numeric difficulty score of 1, 5, and 9, from easiest to hardest. The difficulty score is therefore the average difficulty score across all task instances.

Figure 15 summarizes our findings for difficulties across different SWE-bench style datasets. We provide a more thorough rundown of task instances per difficulty level in Table 14. We find that different SWE-smith bug generation methods yield different levels of difficulty. LM Modify are consistently rated to be easy - from several manual spot checks, we notice that while the prompt for LM Modify provides several examples of types of bugs and does not name specific issues to create, the large majority of bugs created by this strategy are simple variable assignment mistakes (e.g. `a=a; b=b` is changed to `a=b; b=a`). An open-ended prompt like ours does not actually yield high diversity in terms of mistakes created. Procedural modifications are, as expected, the next easiest, as the types of bugs created by this strategy are finite. PR Mirrors and LM Rewrites yield much harder tasks,

Table 14: The score is averaged over all task instances, where easy/med/hard corresponds to 1/5/9. For SWE-smith, we sample 1000 task instances per bug strategy.

| Dataset | # Instances | Score | easy | med | hard |
|---|---|---|---|---|---|
| SWE-bench | 2294 | 5.014 | 438 | 1408 | 446 |
|     Lite | 300 | 3.893 | 93 | 197 | 10 |
|     Verified | 500 | 3.960 | 173 | 284 | 43 |
| SWE-bench Multimodal | 510 | 6.036 | 55 | 265 | 186 |
| SWE-gym | 2438 | 5.625 | 288 | 1456 | 664 |
|     Lite | 230 | 3.890 | 67 | 156 | 4 |
| SWE-smith (LM Modify) | 1000 | 3.304 | 441 | 542 | 17 |
| SWE-smith (LM Rewrite) | 1000 | 5.272 | 68 | 796 | 136 |
| SWE-smith (Procedural) | 1000 | 3.596 | 374 | 603 | 23 |
| SWE-smith (PR Mirror) | 1000 | 4.876 | 206 | 619 | 175 |
| SWE-smith (Combine) | 1000 | 5.720 | 52 | 716 | 232 |

confirmed not only by our bug rating model, but also the lower average resolve rate on these tasks by our expert model (SWE-agent + Claude 3.7 Sonnet). Finally, aggregating smaller functions together is a simple but effective strategy for creating bugs that are rated as more complex. This effect aligns with our original expectations; generally, bugs that require editing more functions and files tend to be rated as more difficult. SWE-smith can be used to create task instances with a range of difficulties.

# F    Experiments

In this section, we provide additional details about the configurations and parameters used to generate trajectories with an expert model and run inference on a fine-tuned model. We then provide additional ablations and analyses about the SWE-smith dataset and the agents trained on SWE-smith.

## F.1    Training Details

**Rejection sampling fine-tuning.** Our fine-tuning setup heavily inherits from Pan et al. [31]'s work. We perform full parameter fine tuning using the `torchtune` [32] library, with learning rate `5e-5`, maximum 3 epochs, and max context length of 32768. Training was carried on Modal [23] on 2-8 NVIDIA H100 80G GPUs. As discussed in Section 3, the procedure for rejection sampling fine-tuning (RFT) is as follows. We first generate expert demonstrations/trajectories using SWE-agent and a "strong" model (e.g. Claude 3.7 Sonnet, GPT 4o) on SWE-smith task instances. Of these, we then only train a student model on the trajectories corresponding to resolved instances.

---

**Task Instance Prompt provided to SWE-agent**

<uploaded_files>
{{working_dir}}
</uploaded_files>
I've uploaded a python code repository in the directory {{working_dir}}. Consider the following PR description:

<pr_description>
{{problem_statement}}
</pr_description>

Can you help me implement the necessary changes to the repository so that the requirements specified in the <pr_description> are met? I've already taken care of all changes to any of the test files described in the <pr_description>. This means you DON'T have to modify the testing logic or any of the tests in any way! Your task is to make the minimal changes to non-tests files in the {{working_dir}} directory to ensure the <pr_description> is satisfied. Follow these steps to resolve the issue:
1. As a first step, it might be a good idea to find and read code relevant to the <pr_description>
2. Create a script to reproduce the error and execute it with 'python <filename.py>' using the bash tool, to confirm the error
3. Edit the source code of the repo to resolve the issue
4. Rerun your reproduce script and confirm that the error is fixed!
5. Think about edgecases and make sure your fix handles them as well Your thinking should be thorough and so it's fine if it's very long.

---

**SWE-agent configuration.** We use two different configurations, one for generating trajectories with an expert model, and a separate one for running inference on the fine-tuned Qwen, student models. The configurations are generally quite similar, with minor differences around how LMs' responses are elicited, the parsing mechanism for an LM response, constraints around message sizes, and the system prompt.

We will first review the information common to both configurations. The prompt template informing an agent of the task's nature and problem statement is included in Figure F.1. This prompt is very similar to the original SWE-agent prompt used in Yang et al. [49]. The prompt templates for showing environment feedback are identical as well. If there is execution output, the text is simply preceded by `OBSERVATION: [output]`. If there is no output (e.g `rm -r` succeeds silently), then the agent is

informed "Your command ran successfully and did not produce any output". The agent computer interface (ACI) provided is also identical; SWE-agent provides LM with access to three general tools:

- `bash`: Execute a bash command in terminal.
- `str_replace_editor`: A tool for viewing, creating, and editing files.
- `submit`: A special keyword for the LM to indicate the task is completed or if it is unable to proceed further with the task.

We briefly review the distinctions. First, tool invocation works differently for expert versus student models. For the Claude and GPT series models that are used as experts, we use function calling for models to invoke the aforementioned tools. On the other hand, the student model is asked to generate a response with XML tags to delineate the thought and action. Therefore, when fine-tuning on expert trajectories, a key processing step is to convert the expert trajectories' function calling format into the XML style response — fine-tuning *directly* on the expert trajectories does not work.

We note that we use these particular settings because as of the publication of this paper, this tool setting reflects the absolute state-of-the-art performance achieved with an open source agent system (SWE-agent) and any existing LM (Claude 3.7 Sonnet). It is certainly possible to explore more tool designs and experiment with different formatting calls, as many existing prior works, notably Yang et al. [49], have performed. However, given the focus of our work, we do not bother with repeating such a "hyperparameter sweep" across configurations for the agent system, as this effort is expensive and has already been performed to suggest that the configuration we are using is ideal for expert level performance.

For generating trajectories with expert models, we run with a maximum of 75 steps and a cost limit of $2.00. A run terminates automatically when either of these limits are reached or the context window of the expert model is exceeded. The overwhelming majority of automatic terminations are due to the 75 maximum steps limit.

For running inference with student models, we run with a maximum of 75 steps or a cost limit[3] of $2.00, where the run similarly terminates when either the steps, cost or context window limit is reached. For the student model, per LM inference call, we truncate the message history to only keep the 5 most recent tool outputs. While we occasionally sample trajectories with the expert model set at various temperatures, for the student model, the temperature is fixed at 0.0.

### F.2  Evaluation Datasets

**SWE-bench.** SWE-bench is a widely used benchmark that evaluates AI systems on their ability to resolve GitHub issues [18]. Given a codebase along with a description of a bug or feature, the AI system is asked to modify the codebase in such a way that the issue presented in the description is resolved. SWE-bench consists of 2294 such task instances, collected from real world pull requests (PRs) and issues in 12 GitHub repositories that are predominantly Python. As discussed in Section 3, the Lite and Verified subsets are curated from the main SWE-bench repository with the goal of making evaluation either more efficent or more reliable. Since evaluation on the entirety of SWE-bench is fairly costly and does not have as many comparable references, we do not evaluate `SWE-agent-LM-32B` on the entire SWE-bench test set.

**SWE-bench Multimodal.** SWE-bench Multimodal applies SWE-bench collection strategy to 12 additional predominantly JavaScript and TypeScript GitHub repositories, where task instances are associated with issues that have visual asset(s) in them [50]. The evaluation dataset consists of 510 task instances. While the original work evaluates vision language models (VLMs) specifically, we do not evaluate `SWE-agent-LM-32B` which, as it is based on Qwen 2.5 Coder Instruct, does not have the ability to process images as inputs.

**SWE-bench Multilingual.** SWE-bench Multilingual is an evaluation dataset consisting of 300 task instances that we introduce with this work. A single author carried out SWE-bench's collection strategy for 42 additional GitHub repositories, covering the following 9 programming languages: JavaScript, TypeScript, C, C++, Go, Java, PHP, Ruby, and Rust. These repositories span a wide range of application domains, including web frameworks, data storage and processing tools, core utilities, and widely used libraries. A brief summary of the dataset is presented in Table 15.

---

[3]We include the cost limit in addition the step limit to provide realistic behavior with respect to handling long context. To calculate a cost value for our model, we use the gpt-4o cost function as of April, 2025.

Table 15: Number of task instances per repository and language in the SWE-bench Multilingual evaluation set. The entire dataset includes 300 task instances covering 9 languages.

| | |
|---|---|
| jqlang/jq | 9 |
| redis/redis | 12 |
| micropython/micropython | 5 |
| valkey-io/valkey | 4 |
| nlohmann/json | 1 |
| fmtlib/fmt | 11 |
| **C/C++** | **42** |
| prometheus/prometheus | 8 |
| caddyserver/caddy | 14 |
| gin-gonic/gin | 8 |
| hashicorp/terraform | 5 |
| gohugoio/hugo | 7 |
| **Go** | **42** |
| briannesbitt/carbon | 10 |
| laravel/framework | 13 |
| phpoffice/phpspreadsheet | 10 |
| php-cs-fixer/php-cs-fixer | 10 |
| **PHP** | **43** |

| | |
|---|---|
| apache/druid | 5 |
| reactivex/rxjava | 1 |
| apache/lucene | 9 |
| projectlombok/lombok | 17 |
| google/gson | 9 |
| javaparser/javaparser | 2 |
| **Java** | **43** |
| babel/babel | 5 |
| mrdoob/three.js | 3 |
| vuejs/core | 5 |
| preactjs/preact | 17 |
| axios/axios | 6 |
| immutable-js/immutable-js | 2 |
| facebook/docusaurus | 5 |
| **JS/TS** | **43** |

| | |
|---|---|
| rubocop/rubocop | 16 |
| jekyll/jekyll | 5 |
| faker-ruby/faker | 2 |
| fastlane/fastlane | 7 |
| fluent/fluentd | 12 |
| jordansissel/fpm | 2 |
| **Ruby** | **44** |
| tokio-rs/axum | 7 |
| nushell/nushell | 5 |
| sharkdp/bat | 8 |
| burntsushi/ripgrep | 2 |
| uutils/coreutils | 5 |
| tokio-rs/tokio | 9 |
| astral-sh/ruff | 7 |
| **Rust** | **43** |

Like SWE-bench Verified, we curate the dataset by excluding task instances deemed by a team of three authors to have ambiguous or underspecified issue text. Each task instance edits (meaning additions and removals) on average 48 lines of code. Similar to SWE-bench and SWE-smith, the median number of Fail-to-Pass tests is one.

We introduce SWE-bench Multilingual to:

1. Provide a benchmark to evaluate model and agent performance across a variety of programming languages and application domains. Existing agent systems often rely on Python-specific tooling, effectively overfitting to the original SWE-bench [50]. Although SWE-bench Multimodal addresses this to some degree, its focus on visual inputs is a confounding factor for text-only evaluation of software engineering capabilities.

2. Remain fully compatible with SWE-bench, so current users can adopt it without changing infrastructure.

3. Keep the dataset small enough to run quickly. While concurrent work like Zan et al. [55] provides more task instances in multiple languages, we purposely constrain the number of task instances so that the dataset is easy to run quickly.

In §F.4, we briefly discuss how performance by existing state of the art methods for SWE-bench is markedly worse on SWE-bench Multilingual, then offer some clear directions for potential next steps to build better agentic coding models that would involve extending SWE-smith.

### F.3 Trajectory Dataset Breakdown

We provide a thorough review of the dataset of SWE-agent trajectories released with this work in Table 16. The majority are generated with `claude-3-7-sonnet-20250219`. To compare with prior work, a minority were generated with `claude-3-5-sonnet-20240620` and `gpt-4o-2024-08-06`. As mentioned in Section 4, to guard against the easy data bias phenomenon, we impose a per-instance cap of 3, meaning for any task instance, we include at most 3 trajectories successfully resolving that task instance in our fine-tuning dataset. From the pool of trajectories reflected in Table 16, we curate a set of 5000 trajectories that we then use to train `SWE-agent-LM-32B`.

Tables 17 and 18 show what repositories and bug types are represented in the final training dataset. In total, 123 repositories are represented, with at least 10 trajectories from 91 repositories. Trajectories

Table 16: Breakdown of trajectories sampled from SWE-smith. Trajectories were generated from subsets of SWE-smith that were either for the purpose of ablations or performance. All trajectories were generated with a maximum of 75 steps and a $2 cost limit.

| Purpose | Bug Gen. | Issue Gen. | # Instances | Temp. | # Traj. |
|---------|----------|------------|-------------|-------|---------|
| `claude-3-7-sonnet-20250219` | | | | | |
| Ablation (Bug Type) | LM (Modify) | LM | 1000 | 0 | 605 |
| | LM (Rewrite) | LM | 1000 | 0 | 507 |
| | Procedural | LM | 1000 | 0 | 745 |
| | PR Mirrors | LM | 1000 | 0 | 557 |
| Ablation (Issue Type) | PR Mirrors | Fixed | 600 | 0 | 259 |
| | PR Mirrors | F2P Test | 600 | 0 | 390 |
| | PR Mirrors | Original | 600 | 0 | 328 |
| | PR Mirrors | LM | 600 | 0 | 319 |
| Ablation (Repositories) | Procedural | LM | 1000 | 0 | 721 |
| | Procedural | LM | 1000 | 0 | 709 |
| | Procedural | LM | 1000 | 0 | 723 |
| | Procedural | LM | 1000 | 0 | 707 |
| Final Dataset Curation | LM (Rewrite) | LM | 3574 | 0 | 1003 |
| | PR Mirrors | LM | 1049 | 0 | 349 |
| `claude-3-5-sonnet-20250219` | | | | | |
| Compare with prior work | All | LM | 800 | 0 | 535 |
| `gpt-4o-2024-08-06` | | | | | |
| Compare with prior work | All | LM | 200 | 0 | 89 |

Table 17: Bug types represented in final training dataset.

| Bug Type | Count |
|----------|-------|
| Combine (File) | 123 |
| Combine (Module) | 7 |
| LM (Modify) | 11 |
| LM (Rewrite) | 1532 |
| Procedural | 1495 |
| PR Mirror | 1848 |

Table 18: Top ten repositories by number of trajectories represented in final dataset for main result.

| Repository | Count | Repository | Count |
|------------|-------|------------|-------|
| getmoto/moto | 378 | sqlfluff/sqlfluff | 122 |
| pandas-dev/pandas | 320 | pylint-dev/astroid | 110 |
| conan-io/conan | 243 | pydicom/pydicom | 103 |
| pydantic/pydantic | 209 | tobymao/sqlglot | 101 |
| iterative/dvc | 181 | pygments/pygments | 99 |
| dask/dask | 139 | scanny/python-pptx | 98 |

are on average 58 turns long, meaning an LM typically takes 29 actions for a given demonstration trajectory. We visualize this distribution in Figure 16.

## F.4 Training Analyses

We provide additional experiments and discussions around training `SWE-agent-LM-32B`.

**Pass@k trend line.** To calculate the Pass@1 score discussed in our main result, we ran SWE-agent with `SWE-agent-LM-32B` six times. In Figure 17, we observe increasing performance at higher values of k, a phenomenon that reflects observations in prior works across LMs for software engineering, code generation, web navigation, and theorem proving. While we do not explore work around inference time scaling and training a separate verifier model to select the best solution candidate generated by multiple roll-outs, as done in Pan et al. [31] and Jain et al. [16], `SWE-agent-LM-32B` is fully compatible with the generate-then-select pipelines explored by such works. Given its strong Pass@1 performance, `SWE-agent-LM-32B` would likely be quite competitive for Best@k results as well. As mentioned before, all trajectories generated in the course of SWE-smith have been released publicly, which the community might find useful for training better verifiers.

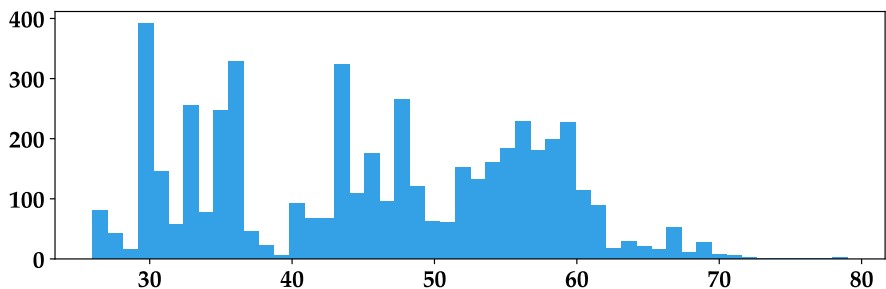

Figure 16: Distribution of number of turns for trajectories represented in the final dataset.

**Rejection sampling fine-tuning ablation.** To confirm that rejection sampling fine-tuning leads to better performance on the downstream task, we compare against a setting where we randomly sample n training points with no filtering criteria, at `n = [100, 200, 400, 800, 1600]` and fine-tune the same student model (Qwen 2.5 Coder Instruct 32B. We then run SWE-agent with each student model on the SWE-bench Verified dataset three times, with the "% Resolved" corresponding to the Pass@1 score. We show results in Figure 18, which confirms that fine-tuning only on trajectories corresponding to successfully resolved tasks is better than randomly sampling trajectories.

**SWE-bench Multilingual performance.** To assess how well `SWE-agent-LM-32B` and existing models generalize to non-Python coding domains, we evaluate the performance of our model, Qwen 2.5 Coder Instruct 32B, and Claude 3.7 Sonnet with SWE-agent on our new dataset, which we introduced in Section F.2. Out of 300 task instances, we found that Claude 3.7 Sonnet achieved a 43% Pass@1 resolve rate, which is significantly better than `SWE-agent-LM-32B` (8.4%) and Qwen 2.5 Coder Instruct (6.5%). `SWE-agent-LM-32B` does not demonstrate a significant improvement over the baseline model. Through several spot checks of different trajectories, we came to a working hypothesis that while the rejection sampling fine-tuning process had improved its ability to carry out multi-turn interactions in this task setting, there were instances where code edits reflected syntax closer to Python despite code and files viewed in previous steps clearly not being written in Python.

While the result for `SWE-agent-LM-32B` SWE-bench Multilingual is clearly subpar, we are excited by such a finding, as it motivates future work on top of SWE-smith. To elaborate, we expect that the path to open agent coding models capable of generalizing to many repositories and languages will be paved by more data and better training techniques, both of which SWE-smith is very capable of facilitating. First, regarding data, although we wrote SWE-smith to be Python centric, the collection methodology and bug generation techniques (especially LM based methods) should be readily transferable to other repositories. Second, the negative result on SWE-bench Multilingual provides a clear impetus for exploring whether better training techniques could lead to models that are trained on one code domain (e.g., Python), but can generalize to many languages and repositories.

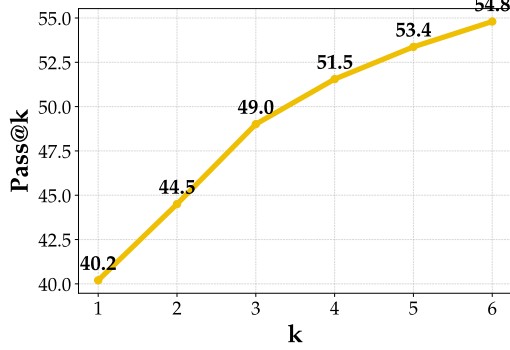
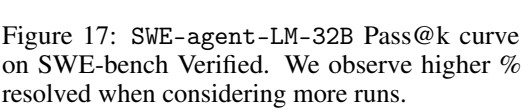
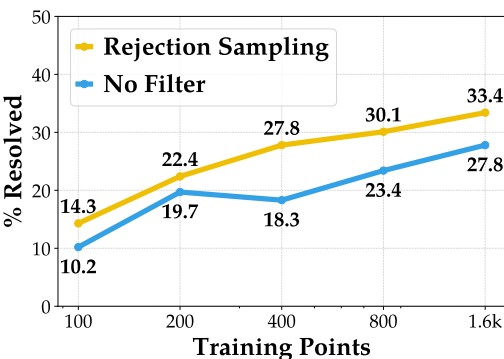

Figure 17: `SWE-agent-LM-32B` Pass@k curve on SWE-bench Verified. We observe higher % resolved when considering more runs.

Figure 18: We confirm that rejection sampling fine-tuning leads to better performance than random sampling of trajectories for training.

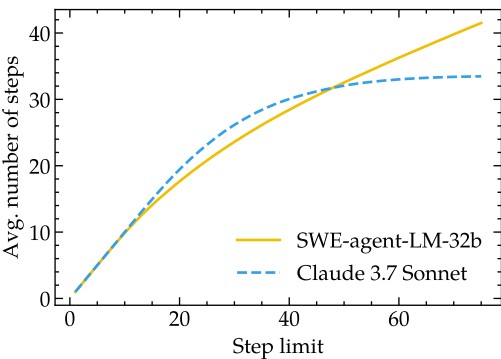
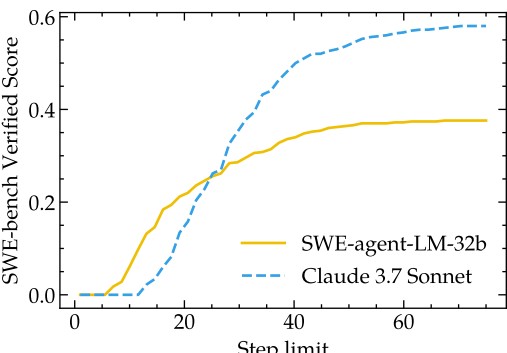

Figure 19: The average step count depends strongly on the prescribed step limit.

Figure 20: Number of successful instances submitted before a given step limit.

### F.5 Agent Behavioral Studies

#### F.5.1 Turn counts and cost

While agents are frequently quoted with a singular cost-per-instance number, this can be very misleading in the case of SWE-agent-LM-32B. Because most of the failed instances fail due to termination by the cost or turn count limit, the average cost and turn counts depend strongly on these limits (see Fig. 19).

We can also chart the number of resolved instances vs step limits. To avoid reevaluating the agent with multiple step limits, we use one run with step limit 75 and then assume that a successful agent run that terminates after step $n$ would have failed when restricted by a limit smaller than $n$. This chart corroborates the point made in section 3: SWE-agent-LM-32B has a higher resolution rate for very low step limits.

#### F.5.2 Analysis of agent action space

**Reduction to *base commands*.** In addition to the dedicated tools provided to the agent as part of the agent computer interface (Section F.1), the agent can execute arbitrary bash commands. This makes quantitative analyses of the agent action space challenging. For example, the agent might issue commands like `PYTHONPATH=/testbed/repo cd /testbed/repo && python3 reproduce.py`. We have found the following procedure to determine a *base command* effective to meaningfully describe the action:

1. Strip any environment variable manipulation from the beginning of the command.

2. When multiple commands are chained with `&&` or semicolons, only consider the last command.

3. Remove all arguments. Because some commands have subcommands (e.g., `git checkout`), we apply several basic heuristics to determine whether to keep the first or the first two words.

**Repetitive actions.** We determine the longest repetitive sequence of actions by determining the longest sequence of identical base commands within the agent actions. Note that this means that e.g., `str_replace_editor view` actions that target different files are considered to be repetitive actions as far as this analysis is concerned.

#### F.5.3 Failure mode analysis

Categorizing the failure mode proceeds as shown in Figure 21:

1. **Error conditions:** If the agent terminates due to an error (environment errors, inability of the LM to correctly format its messages, etc.) or because it exceeded its maximum context window, we return the **error** or **context** category.

2. **Early termination:** If the agent was terminated because of a step or cost limit, we return one of the **stuck ...** subcategories. Note that the SWE-agent still attempts to extract a

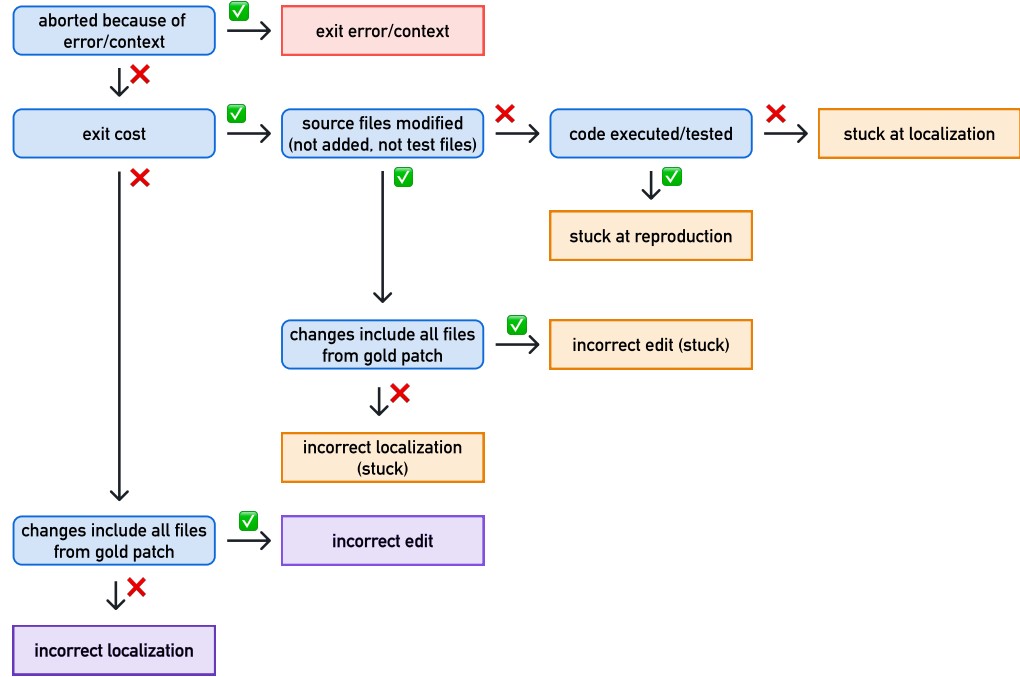

Figure 21: Categorizing failure modes

submission (list of changes/patch). We determine the subcategory based on which part of the workflow agentic loop was terminated:

(a) If no source (i.e., non-test) file was modified[4] and no attempt at testing was made, we return **stuck at localization**. If test commands were run (i.e., `python`, `pytest`, . . . , or similar commands), we return **stuck at reproduction**.

(b) If source files *were* modified, we check whether the changes include changes to all source files that are modified in the gold patch. If not, we return **incorrect localization (stuck)**, else **incorrect edit (stuck)**.

3. **Successful submission:** If the agent terminated and submitted a solution naturally, we return **incorrect localization** or **incorrect edit**, depending on whether the changes from the submitted patch included changes to all files from the SWE-bench gold patch.

### F.5.4  Mitigating repetitive actions

As described in section 4.2, `SWE-agent-LM-32B` frequently shows highly repetitive actions for unresolved instances. In light of this, it seems promising to investigate whether agent scaffolding interventions can be used to mitigate the problem and increase the success rates.

We make the following modification to the agent scaffold:

- We add warning messages to the observation (command output) if a base command is repeated four (`str_replace_editor view`) or six (any other base command) times. The warning message advises to try different commands, and in particular suggest to locate relevant context using `find` or `grep`.

- If the warning messages do not break the string of repetitive base commands and the repetition length reaches 6 (`str_replace_editor view`) or 8 (any other base command), every following action is resampled up to 10 times, stopping at the first base command that is distinct from the previous ones. To further increase the likelihood of breaking the cycle, we inject assistant messages or raise the temperature if the repetition length reaches 7 or 9.

---

[4]We exclude added files because solving SWE-bench instances always requires *changes* to existing files.

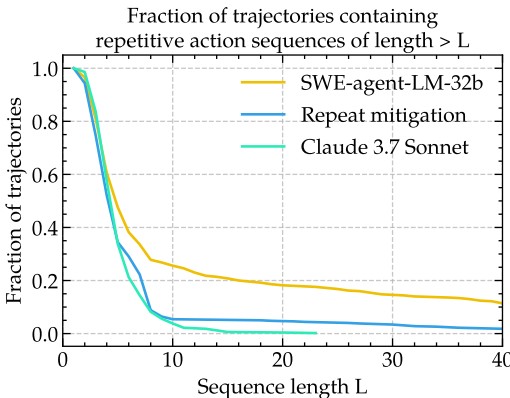

Figure 22: Scaffold interventions can drastically reduce the number of repetitive actions.

This effectively reduces the number of repetitive actions (see Fig. 22). However, the overall number of resolved instances drops slightly to 192 (38.4%). Variations of the above strategies yield similar outcomes: while repetition is suppressed, success rates do not improve substantially. This may suggest that repetitive actions are better understood as *symptoms* of the model's difficulty in solving an instance (such as when the instance is out-of-distribution or particularly challenging) rather than constituting intrinsic failure modes.

## G   Miscellaneous

**Teaser figure description.** We briefly describe how the left hand graph of Figure 1, which depicts scaling of task instance collection for the SWE-smith vs. SWE-bench, was created. For SWE-smith, we simply collected the number of task instances for each repository. For SWE-bench, we ran the SWE-bench task instance candidate collection script on all 128 repositories, which first crawls all PRs from a given repository. Then, each PR that edits at least one or more Python files and changes at least one or more testing related files is converted into a candidate task instance. Finally, based on the average task instance yield rate reported in Jimenez et al. [18], we estimate the number of viable task instances to be 20% of the candidates. We then determine the number of task instances for n repositories at intervals of 5 repositories ranging from 5 to 250, where the repositories are sorted by number of stars. In other words, the first five repositories we account for in the figure are the five with the fewest number of stars out of the 128 repositories used.

**Extended related works.** We discuss additional related works briefly, primarily about similar work towards synthesizing trajectories for training LM agents, but for the domain of web tasks. To improve the interactive capabilities of open source LMs [7], prior works have also explored trajectory generation techniques for web benchmarks and settings [45, 51, 59]. For web navigation, existing strategies rely on (1) performing random walks which are then labeled retroactively with instructions [43, 25], (2) using online web tutorials as a source of indirect supervision for generating synthetic trajectories [30], or (3) collecting human demonstrations [34, 47]. These procedures do not translate well to the software engineering setting; random sequences of command line interactions usually do not achieve meaningful effects on a codebase. Our cursory efforts around replaying trajectories synthesized from online code edit sequences (e.g. GitHub commit histories) were unsuccessful due to the limited information available, which primarily capture file-level changes without reflecting the underlying skills, decision-making, or the broader context of a software development process.

Our exploration of using SWE-agent to automatically determine installation and testing specifications for a repository is heavily influenced by two research directions - automatic execution environment construction using LMs [6, 11, 37], and generating unit tests using LMs [26]. Although relatively much less than SWE-bench style collection, SWE-smith still requires minimal amounts of human labor (around 8 minutes total per repository). As we expand SWE-smith to more repositories and languages, we are continuing to consider how to completely automate the environment construction process end to end.

**Societal Impacts.** While SWE-smith introduces powerful capabilities for generating software engineering training data at scale, we are mindful of its potential societal impacts and have taken steps to mitigate associated risks. For example, although automated bug generation could in principle be misused to introduce vulnerabilities, our system is explicitly designed to operate in isolated, sandboxed environments. We do not create or submit any artifacts to real-world repositories, ensuring that our methods do not interfere with ongoing development or introduce instability into open-source projects. All bug generation techniques are well documented, and we release data only after rigorous validation, including filtering for test-breaking changes and confirming reproducibility. We also address fairness and privacy concerns by targeting a diverse set of repositories and avoiding the collection or use of contributor metadata. While systems trained on SWE-smith may produce incorrect outputs even when functioning as intended, we encourage human-in-the-loop workflows and provide infrastructure for auditability and monitoring. On the positive side, SWE-smith significantly lowers the barrier for building open-source software engineering agents, enabling broader participation in research and innovation. It supports advances in program repair, debugging, and education, and facilitates the study of agent behavior in realistic settings—ultimately helping make software systems more reliable and accessible.

