# OpenReview forum: "SWE-smith: Scaling Data for Software Engineering Agents"
_NeurIPS.cc/2025/Datasets_and_Benchmarks_Track — NeurIPS 2025 Datasets and Benchmarks Track spotlight_

### Official Review · Reviewer_4V7Q · 2025-06-22

**Rating:** 5
**Confidence:** 5

**Summary:**

The paper introduces SWE-smith, an automated pipeline that scales the construction of executable, test-validated software-engineering tasks from tens of repositories to 50 k instances while reducing storage and human cost by ~500×. It publicly releases the dataset, code and a 32 B open-source model that achieves a new state-of-the-art 40.2 % Pass@1 on SWE-bench Verified.

**Dataset Code Accessibility:**

Yes

**Dataset Code Comments:**

The dataset is fully accessible in its final form via the Hugging Face platform. The authors provide Python code snippets for both the datasets and mlcroissant libraries, enabling reproducibility and ease of use.  Although one resource file is marked inaccessible, it does not affect the overall usability of the dataset. The submission demonstrates a strong commitment to reproducibility and openness.

**Ethical Considerations:**

No, there are no or only very minor ethics concerns

**Final Justification:**

Thank you for the rebuttal. I like this job due to its soundness. I will maintain my score.

**Limitations Weaknesses:**

1. Language scope: The current pipeline targets Python only; extending the AST tooling and environment builder to other major languages (Java, JS, C++) would broaden applicability.
2. Training method: Results rely solely on supervised fine-tuning. Incorporating reinforcement learning may improve the agent’s robustness and step efficiency.
3. Synthetic-bias assessment: Although diverse, the majority of bugs are artificial. A deeper study on how synthetic distributions align with real-world bug patterns and their impact on generalisation would strengthen the empirical claims.

**Strengths Contributions:**

1. The paper introduces a highly scalable data-generation pipeline whose “build-once, mutate-many” strategy reduces storage and human cost by roughly 500× relative to per-task containers, making it the first practical approach to synthesize tens of thousands of executable software-engineering tasks from real repositories.
2. This method is paired with a comprehensive public release—50 k tasks, their Docker environments, generation scripts, and a 32 B fine-tuned model—thereby filling a critical resource gap and enabling fully reproducible SE-agent studies.
3. On the public benchmark, SWE-bench Verified, the resulting model surpasses the 40 % Pass@1 threshold and narrows the gap to GPT-4o, proving that data scale rather than proprietary architecture alone can drive significant performance gains.

---

> ### Author Rebuttal · Authors · 2025-07-31
>
> # Response to Reviewer 4V7Q
>
> We thank Reviewer 4V7Q for highlighting SWE-smith’s highly scalable “build-once, mutate-many” pipeline and open-source contributions, and for the insightful questions about language generalization, RL-based training, and assessing synthetic-bug bias.
>
> [Link to General Responses (Included with response to Reviewer PUO9)](https://openreview.net/forum?id=63iVrXc8cC&noteId=Rpf3zZjbdn)
>
> ---
>
> > W1: Language scope - The current pipeline targets Python only; extending the AST tooling and environment builder to other major languages (Java, JS, C++) would broaden applicability.
>
> **“General Response #1” should address these concerns.** We definitely agree that expanding to other languages makes the SWE-smith toolkit more impactful, and our response shows how we are pursuing this direction for SWE-smith. .
>
> ---
>
> > W2: Training method - Results rely solely on supervised fine-tuning. Incorporating reinforcement learning may improve the agent’s robustness and step efficiency.
>
> **“General Response #2” should address these concerns.**
> We very much agree with your assessment that RL may address some, if not many issues with agent behavior we identified, like repetitive actions.
> However, as concurrent work suggests, RL training requires significant resources and is orthogonal to evaluating our primary hypotheses.
> For this reason, we’ve decided to leave exploring RL with SWE-smith for future work.
>
> ---
>
> > W3: Synthetic-bias assessment - Although diverse, the majority of bugs are artificial. A deeper study on how synthetic distributions align with real-world bug patterns and their impact on generalisation would strengthen the empirical claims.
>
> We thank the reviewer for raising this point — assessing the viability of synthetic bugs as training data is central to SWE-smith’s contribution.
> Our paper contains several analyses that aim to compare synthetic task instances to real-world bugs both empirically and structurally.
>
> **We have several sections discussing how SWE-smith and synthetic bugs compare to real world PRs as training instances.**
> * Empirical Comparisons (Section 4):
>     * _Bug Type vs. Performance (§4.1, Table 4)_: We compare models fine-tuned on trajectories from different bug types. “PR Mirror” reflects real-world PRs. LM Rewrite (8.8%) and Procedural Modification (8.6%) yield comparable pass@1 scores to PR Mirror (9.2%) on SWE-bench Verified.
>     * _SWE-smith vs. Real-PR Dataset (Lines 187–191)_: With the same number of training trajectories (500), SWE-smith outperforms SWE-gym (real-world PRs from 11 repos) by +8.2%, using the same base model and expert.
> * Structural Comparisons (Appendix D, Figure 14):
>     * We compare surface-level features (e.g., lines/files edited, F2P tests) between SWE-smith and prior real-world datasets (SWE-bench, SWE-gym).
>     * SWE-smith task instances fall between SWE-bench Verified and SWE-bench/SWE-gym in edit size.
>     * SWE-smith’s F2P test distribution is more balanced — 67% of tasks break 2+ tests, compared to ~30% for SWE-bench Verified.
> * Difficulty Ratings (§2.2, Appendix E):
>     * Using a learned difficulty model trained on human-labeled SWE-bench data, we find that SWE-smith covers a comparable difficulty range.
>     * Different bug generation strategies yield distinct difficulty profiles (e.g., LM Rewrite and PR Mirror tend to be harder).
>
> **Based on your suggestions, we’ve added new analyses in Appendix D to examine synthetic vs. real bugs further.**
> * We visualize CDFs for # lines edited and # F2P tests for synthetic (LM generated, Procedural Mod., Combined) vs. real (PR Mirror) bugs.
>     * Versus LM generated, Procedural Mod. bugs, PR Mirrors require more lines edited, but break fewer unit tests.
>     * Combined bugs break more tests and require lengthier fixes.
> * We will also include a new case study categorizing synthetic (SWE-smith) vs. real (SWE-bench) task instances corresponding to the `sympy/sympy` repository. Given 50 bug patches per set, we prompted Claude 3.7 to briefly describe the bug, then cluster the descriptions to identify general bug categories. This experiment is still running; we will do our best to complete it by the discussion deadline.
>
> We hope these analyses offer a comprehensive examination of synthetic vs. real bugs.
> Deeper alignment studies (e.g., on bug category, localization difficulty, edit intent) are a valuable direction for future work
> If the reviewer has additional concrete suggestions, we’d be happy to consider and incorporate them.

---

### Official Review · Reviewer_dLbQ · 2025-07-02

**Rating:** 5
**Confidence:** 4

**Summary:**

In this paper, the authors introduce SWE-Smith, a novel pipeline for generating software engineering training data at scale. They create a dataset consisting of 50,000 instances sourced from 128 GitHub repositories, which is an order of magnitude larger than all previous works. The authors train the SWE-agent-LM-32B model, achieving a 40.2% Pass\@1 resolution rate on the SWE-bench Verified benchmark, setting a new state of the art among open-source models.

**Additional Feedback:**

1. The training method used in the paper is reject sample fine-tuning. I wonder if some reward-based RL methods could potentially perform better in this environment. Based on previous works, it's not difficult to define the state and action to construct a reinforcement learning environment for these coding tasks. In Section 4.2, the paper mentions the issue of repetitive actions, which I believe is a limitation of the fine-tuning method. Trying methods like GRPO and designing an appropriate reward could possibly improve this issue.

2. The paper uses SWE-Smith to build the data from 128 GitHub repositories. I would like to know if there is a more detailed analysis of the statistics of these repositories, such as the impact of the size of a repository on the agent during training and inference. I think it would be meaningful to compare the training performance between using a few large engineering repositories, like Torch and Transformers, versus using a large number of small repositories to construct the training set.

If you could address these two issues, I would consider increasing the score.

**Dataset Code Accessibility:**

Yes

**Dataset Code Comments:**

The dataset code is accessible.

**Ethical Considerations:**

No, there are no or only very minor ethics concerns

**Final Justification:**

Thank you to the authors for the thorough and comprehensive rebuttal. Since it addressed all of my concerns, I have raised my score.

**Limitations Weaknesses:**

1. The paper evaluates the results solely based on the SWE-bench. Are there additional benchmarks in the SWE field that could be used for evaluation and reference? I understand that SWE-Smith is designed to collect data for general SWE tasks, but providing evaluation results on some coding sub-tasks, such as CodeXGLUE, would give us a more comprehensive understanding of the model.

2. In Table 3, it is mentioned that "All performance numbers are pass\@1. We do not compare against systems that use verifiers or multiple attempts at test time." I question the rationale behind this. Test-time scaling is also a hot research direction in coding tasks, and I would like to see some systems from this area included for comparative evaluation.

**Strengths Contributions:**

The paper proposes a systematic SWE data collection method, using SWE-Smith to collect data from 128 Python repositories, generating a total of 50,000 instances. This collection method covers a variety of SW bug types. Using the data collected with this method, the authors trained the SWE-agent-LM-32B model and tested it on the SWE-bench, achieving state-of-the-art performance in Open Weight Models. Furthermore, the paper provides an in-depth analysis of SWE-Smith’s bug and problem statement generation strategies, as well as Agent Behavior, highlighting areas for future improvements. Overall, this work introduces new methods and approaches to SWE data collection, offering both innovation and insights for future research.

---

> ### Author Rebuttal · Authors · 2025-07-31
>
> # Response to Reviewer dLbQ
>
> We thank Reviewer dLbQ for acknowledging SWE-smith’s large-scale data-generation pipeline and open-weight SOTA results, and for the constructive suggestions on additional benchmarks, test-time scaling, RL fine-tuning, and deeper repository-level analyses.
>
> [Link to General Responses (Included with response to Reviewer PUO9)](https://openreview.net/forum?id=63iVrXc8cC&noteId=Rpf3zZjbdn)
>
> ---
>
> > W1: The paper evaluates the results solely based on the SWE-bench. Are there additional benchmarks in the SWE field that could be used for evaluation and reference?
>
> We thank the reviewer for this suggestion and offer the following findings:
>
> We report new numbers for these benchmarks:
>
> * **Aider Polyglot** (225 exercism coding challenges across C++, Go, Java, JS, Python, and Rust). SWE-agent-LM-32B scores 12.4%, 2.1% higher than the Qwen 2.5 Coder Instruct baseline. Minor adjustments were made to adapt the Aider tool calling format to the XML format used during training of SWE-agent-LM-32B.
> * **HumanEvalFix**: Following the original SWE-agent work, we run SWE-agent with SWE-agent-LM-32B on HumanEvalFix-[python, js, java], an evaluation that involves fixing buggy solutions to HumanEval problems. Performance improves from 52.6% to 59.7% (+7.1%) for Python, but stays about the same for js, java (+1.3%).
>
> We’d also like to highlight **SWE-bench Multilingual**, a new evaluation set of 300 SWE-bench style, issue‑resolution tasks spanning 9 languages (JS/TS, C/C++, Go, Java, PHP, Ruby, Rust), introduced in this work.
> * **Appendix F.2** contains details on how SWE-bench is collected + dataset statistics. The collection procedure is very similar to SWE-bench.
> * **Appendix F.4** summarizes evaluation results. In short, SWE-agent-LM-32B (8.4%) improves over Qwen 2.5 Coder Instruct (6.5%). However, the scores are still far behind SWE-agent + Claude 3.7 Sonnet (43%).
> * This eval set has been used by the recent Kimi K2 (47.3%) and Qwen 3 Coder (54.7%) releases.
>
> We plan to update the paper as follows:
> * Briefly mention Aider Polyglot, HumanEvalFix, and SWE-bench Multilingual results in Section 4.
> * Include full result tables + discussions in Appendix F.4.
>
> These additions now give a more holistic picture of our model’s strengths and weaknesses across both issue‑resolution and standalone code‑generation tasks.
>
> ---
>
> > W2: “...We do not compare against systems that use verifiers or multiple attempts at test time." I question the rationale behind this. Test-time scaling is also a hot research direction in coding tasks…
>
> Test-time scaling is indeed an exciting direction, but is orthogonal to the focus of this work.
> SWE-smith aims to demonstrate how fine-tuning on high-quality trajectory data at scale improves _base_ model performance, and pass@1 is a clean way to evaluate that — without the confounding effects of external verifiers, reranking strategies, or multiple rollouts.
>
> Prior work on SWE-bench (e.g., CodeMonkeys, SWE-gym, R2E-gym) consistently shows that **test-time scaling (e.g., best@k using discriminators) improves performance across models — but does not reverse model rankings**. That is, models with higher pass@1 typically also achieve higher best@k when the same test-time scaling method is applied.
>
> Therefore, we believe **pass@1 is sufficient to validate the impact of SWE-smith training**.
> We also include pass@k trendlines in Figure 17 (Appendix F.4), which can be interpreted as upper bounds for best@k performance. SWE-agent-LM-32B shows superior performance under this lens as well.
>
> ---
>
> > Q1: The training method used in the paper is reject sample fine-tuning. I wonder if some reward-based RL methods could potentially perform better in this environment.
>
> **“General Response #2” should address these concerns.**
> We agree with your insight that RL could very likely reduce some of the bad behaviors we reported in the paper.
> However, getting online RL to work with multi-turn task environments is quite tricky.
> Also, as a D&B benchmarks submission, we elected to focus on the data scaling challenge.
> But we do very much agree that RL + SWE-smith is a sensible next step.
>
> ---
>
> > Q2: The paper uses SWE-Smith to build the data from 128 GitHub repositories. I would like to know if there is a more detailed analysis of the statistics of these repositories.
>
> Along with the main paper content (Section 2), Appendix D contains a number of additional breakdown of the SWE-smith dataset:
> * **Pages 34-38** lists each of the 128 repositories along with their provided GitHub description.
> * **Table 12 (Task Instance Yield Rate), Table 13 (Task Instance Statistics), Figure 14 (CDF comparisons with SWE-bench-esque datasets)** quantify the task instances (although not in a repository-specific manner)
>
> If there are additional breakdowns you’d like to see, we’d be happy to provide them.
>
> With the SWE-smith dataset open-sourced on HuggingFace, gathering more insights on the data should be convenient.
> For instance, the following code will show how many task instances there are per repository:
>
> ```python
> from datasets import load_dataset
> from collections import Counter
> repos = Counter(load_dataset("SWE-bench/SWE-smith", split="train")["repo"])
> ```
>
> Regarding the **comparison between training on trajectories from smaller versus larger repos** - thanks for this suggestion, **we found this idea to be very interesting**. We carried out this experiment at the 7B scale, similar to the existing ablations.
> We fine-tune Qwen 2.5 Coder Instruct on two distinct training sets of 250 trajectories each:
> * They are collected from the 5k trajectories SWE-agent-LM-32B was trained on.
> * To control for bug type, we only use trajectories from LM rewrite bugs.
> * To control for issue type, we only use LM generated issue text.
> * We define a small repository as a codebase with 30 or fewer *.py files (excluding tests). 30 files is roughly the median number of files of the 128 repos.
> * We run inference 3 times and calculate pass@1.
>
> The model SFT’ed on **large** repo trajectories achieved **7.6%** on SWE-bench Verified.
> The model SFT’ed on **small** repo trajectories achieved **6.9%**.
> The reason larger repositories seem to be better can be attributed to having **lengthier file localization steps**.
> Post-discussion period, we will revise the paper to add this experiment to **Appendix F**.

---

> > ### Comment · Reviewer_dLbQ · 2025-08-01
> >
> > Thank you for the thorough and comprehensive rebuttal. Since it addressed all of my concerns, I have raised my score.

---

> > > ### Author Response · Authors · 2025-08-09
> > >
> > > Thanks so much, Reviewer dLbQ — it’s been a pleasure to engage with your thoughtful and actionable feedback during the rebuttal period. Your suggestions directly shaped several of the additions and analyses we included, and we truly appreciate the time you took to think about our work.

---

### Official Review · Reviewer_wjp1 · 2025-07-02

**Rating:** 4
**Confidence:** 4

**Summary:**

This paper introduces SWE-smith, a toolkit & dataset for generating large-scale, execution-validated software engineering tasks by automatically synthesizing bugs and problem statements in Python repos. The authors present a pipeline that improves scalability over prior work by building execution environments first and generating 100s to 1,000s of bug-inducing task instances per repository, using a combination of function rewrites via LMs, AST modifications, PR inversion, and bug combination strategies. SWE-smith yields a dataset of 50,000 instances spanning 128 GitHub repos.

**Dataset Code Accessibility:**

Partly

**Ethical Considerations:**

No, there are no or only very minor ethics concerns

**Final Justification:**

I updated my review according to the communication with the authors.

**Limitations Weaknesses:**

1. The paper proposes several methods for synthesizing buggy tasks, but it is evident that these approaches produce tasks with different levels of difficulty. For example, procedural modifications that disrupt code structures tend to introduce particularly obvious errors that LLMs can fix very easily. Moreover, these types of tasks can be generated in large quantities. I am somewhat concerned that if the dataset contains too many of these simpler patterns, the model might overfit to solving them. Currently, LLMs often achieve a 10–30% improvement (essentially learning to solve these simpler tasks), so this may not be very apparent yet. However, for further performance gain, it is unclear whether this can continue to scale effectively.
2. In Fig. 4, the authors use percentage improvements to report performance on swe-bench verify. However, this benchmark only contains 500 instances, and the authors focus on the SymPy repository (I did not see an explicit explanation for choosing SymPy). Given the small data size, large percentage changes may correspond to differences of only a few individual cases, making the conclusions less solid.
3. The generation of issue/problem statements is automated via prompting. Tab.5 ablates some alternatives, but the qualitative fit and diversity of these statements compared to actual GitHub issue styles is not deeply assessed.
4. When reviewing, I checked data from SWE-smith and observed that the quality of the synthesized tasks was not very high (not as good as presented by the paper). The model often attempted to reproduce the issue descriptions directly, only to find that the actual bugs did not match the synthetic descriptions, requiring the model to search for the real issues on its own.
5. Minor: Python-specific. All data collection, task synthesis, and bug generation (especially AST manipulation) rely on Python tooling. While authors claim transferability to other PLs is possible, the current dataset does not demonstrate this

**Strengths Contributions:**

1. SWE-smith offers a much larger dataset than previous SE agent datasets, while substantially reducing human labor and storage requirements
2. Although combining existing techniques, using LM-based function modification, procedural AST modification, PR inversion, and bug combination enables the creation of diverse bugs.
3. The paper provides detailed analysis include multiple ablations, analysis of agent specialization and failure analysis

---

> ### Author Rebuttal · Authors · 2025-07-31
>
> # Response to Reviewer wjp1
>
> We thank Reviewer wjp1 for highlighting SWE-smith’s scalable bug-generation pipeline, diverse synthesis strategies, and detailed analyses, and for the thoughtful concerns about difficulty balance, evaluation granularity, issue-text realism, and language transferability.
>
> [Link to General Responses (Included with response to Reviewer PUO9)](https://openreview.net/forum?id=63iVrXc8cC&noteId=Rpf3zZjbdn)
>
> ---
>
> > W1: …I am somewhat concerned that if the dataset contains too many of these simpler patterns, the model might overfit to solving them…
>
> We appreciate the reviewer’s concern about over-reliance on simpler bug patterns.
> Empirically, we do not observe evidence of overfitting — scaling to more task instances / trajectories continues to improve performance (Figure 1).
>
> We agree that bug types differ in difficulty — this is why we provide detailed difficulty analysis in Section 2.2 (with further discussion in Appendix E) and report difficulty-aware training outcomes in Section 4.1.
> The key takeaway from these analyses is that SWE-smith is capable of synthesizing task instances spanning a wide range of difficulty.
>
> We note that our final training set of 5k trajectories is generated from a subset of 8.7k task instances.
> We apply both trajectory-based filtering (e.g., limiting to 3 per task) and bug-type balancing (Lines 172-180).
> Indeed, SWE-smith is not intended to be used exhaustively; we don’t run the expert model on all 50k instances.
> A key strength of the dataset is that it enables principled curation of training data — deciding which task instances to train on is itself an interesting and open design choice.
>
> Finally, we note that our ablation (Lines 218-229) also points out how training on trajectories that resolve more difficult task instances do not necessarily yield better models – selecting which trajectories to train on is not as straightforward as difficulty.
> We share the reviewer’s interest in whether future gains might depend more on complex tasks, and view this as a valuable direction for continued work.
>
> ---
>
> > W2: In Fig. 4, the authors use percentage improvements to report performance on swe-bench verify. However, this benchmark only contains 500 instances…
>
> To clarify, SWE-bench Verified is 500 instances. The ablation about repo-specialization (Lines 235-248) is evaluated on a subset of SWE-bench Verified that are (1) from SymPy (2) Created after 1/1/2022 = 22 instances.
>
> _If the reviewer is referring to the Sympy-only evaluation being too small…_
>
> We thank the reviewer for raising this point — we agree that with only 22 instances, percentage differences should be interpreted with appropriate caution. However, **our intention in Fig. 4 is not to claim statistical significance, but to illustrate a trend**: repository-specialized models can yield meaningful gains on their target repo, even when trained on relatively small subsets of data.
>
> We chose SymPy because it is one of the SWE-bench test repos, has a well-maintained test suite, and represents a popular and non-trivial Python codebase. That said, we agree that exploring more repositories in this setting would further strengthen the generality of the claim, and we consider this an important direction for future work.
>
> _If the reviewer is referring to SWE-bench Verified being too small…_
>
> We respectfully disagree that 500 instances is too small to draw meaningful conclusions. This size is **comparable to or larger than** many widely used evaluation benchmarks (e.g., HumanEval, MBPP), and has become a standard evaluation set across recent work in the field. Our reported differences (e.g., 40.2% pass@1 vs. prior SOTA) correspond to substantial gains over strong baselines — not small fluctuations due to noise.
>
> ---
>
> > W3: The generation of issue/problem statements is automated via prompting… the qualitative fit and diversity of these statements compared to actual GitHub issue styles is not deeply assessed.
>
> In Section 4.1, we run an ablation to address this question directly (**Lines 205-217**).
> Given 600 PR Mirror task instances, we compare the performance of the same base LM fine-tuned on:
> * 259 trajectories generated from these task instances with **LM generated** issue text.
> * 259 trajectories generated from these task instances with the **original GitHub** issue text.
>
> The performance difference between the two is miniscule - **7.7% for LM generated text vs. 7.8% for GitHub Issue text** when evaluated on SWE-bench Verified.
>
> While we agree that a deeper qualitative analysis of issue diversity and style could be valuable, we believe this result already provides strong evidence that LM generated issue text is sufficiently realistic for training purposes.
> Exploring finer-grained style metrics or human evaluations of issue realism could be a promising direction for future work.
>
> ---
>
> > W4: I checked data from SWE-smith and observed that the quality of the synthesized tasks was not very high (not as good as presented by the paper)
>
> To clarify, we assume the reviewer is referring to cases where the issue text does not exactly describe the underlying bug, potentially affecting the quality of the 5k trajectories used to train SWE-agent-LM-32B.
>
> We appreciate the reviewer inspecting the trajectories and surfacing this observation.
>
> We agree that, in some cases, the automatically generated issue descriptions may not align perfectly with the ground-truth bug.
> However, we believe this is not only tolerable, but in some cases desirable, for several reasons:
> 1. As discussed in SWE-bench Verified, even in real-world GitHub issues, **bug reports can be incomplete, ambiguous, or partially misleading**.
> 2. The quality of an issue as rated by human annotators has been shown to be **highly subjective** in SWE-bench Verified.
> 3. Our goal is **not to perfectly match the issue and patch**, but to produce tasks that **realistically test an agent’s ability to localize and resolve failures**.
>
> Our generation strategy includes the patch, test logs, and relevant test code, providing sufficient context for the LM to write an issue that conveys the bug meaningfully without revealing the solution.
> As mentioned, **Table 5** shows that models trained on LM-generated issue text perform comparably to those trained on original GitHub issues.
> That said, improving issue-bug alignment remains a valuable direction for future work.
>
> ---
>
> > W5: Python-specific. All data collection, task synthesis, and bug generation (especially AST manipulation) rely on Python tooling.
>
> **“General Response #1” should address these concerns.**
> We believe the new artifacts demonstrate that, with minor adaptation, the SWE-smith bug generation pipelines work for non-Python programming languages as well.

---

> > ### Comment · Reviewer_wjp1 · 2025-08-05
> > **Response from Reviewer**
> >
> > Thank you for your response. I would like to follow up on the data quality issue I previously raised. The paper devotes considerable effort to discussing and optimizing the generation of problem statements, as this component directly impacts the overall task quality and the performance of swe agents.
> >
> > That said, based on my review of (multiple versions of) the attached dataset (which appears to have been updated since the NeurIPS submission deadline), I have observed that many data entries still appear to be missing the problem_statement field. For example: https://huggingface.co/datasets/SWE-bench/SWE-smith/viewer/default/train?p=1
> >
> > I am unsure whether this omission is intentional or if there are other considerations the authors have in mind.

---

> > > ### Author Response · Authors · 2025-08-06
> > > **Response to Reviewer wjp1**
> > >
> > > Hi reviewer `wjp1` thanks for highlighting this, we appreciate your thoroughness and agree - the dataset, as is, with missing problem statements limits its utility.
> > >
> > > **Generating issue texts for the remaining ~40k task instances is currently underway and will be pushed to the dataset** by the time paper decisions are announced.
> > >
> > > Before, we didn't have problem statements for the remaining 40k instances because of budget. Generating issue text for a task instance takes $0.0254 on average (Section 2.2), which requires an estimated 1k in cost that we allocated towards generating more task instances and training runs instead.
> > >
> > > Thankfully, we've gotten some [similar feedback](https://github.com/SWE-bench/SWE-smith/issues/7) (also [here](https://github.com/SWE-bench/SWE-smith/issues/61), [here](https://github.com/SWE-bench/SWE-smith/issues/9)) last month, and we're happy to share our efforts that will imminently address this lack of issue text.
> > > * The `CONTRIBUTING.md` now has very simple instructions to [generate issue text](https://github.com/SWE-bench/SWE-smith/blob/main/CONTRIBUTING.md#add-problem-statements).
> > > * Running SWE-smith style issue generation on the remaining 40k task instances is underway. Progress is being tracked in an open-invite [Slack channel](https://join.slack.com/t/swe-bench/shared_invite/zt-36pj9bu5s-o3_yXPZbaH2wVnxnss1EkQ).
> > > * We received a [contribution](https://github.com/SWE-bench/SWE-smith/issues/127) that provides R2E-gym style backtranslations for the remaining 40k instances. We don't plan to incorporate this in the main dataset (the form factor of R2E-gym style issue text is quite different from SWE-smith's), but we will make it available as a separate HuggingFace dataset.
> > >
> > > In short, this omission is original due to cost and will certainly be resolved within the next month.

---

> > > > ### Comment · Reviewer_wjp1 · 2025-08-07
> > > > **Response from reviewer**
> > > >
> > > > Thank you for your response. I believe that for a contribution centered on a dataset and benchmark, the soundness of the data itself is crucial. I hope you can continue to improve the dataset in your revision. I have updated my rating according to our communication.

---

> ### Comment · Area_Chair_7QCz · 2025-08-03
>
> Dear reviewer,
>
> Please read the rebuttal and provide your final justification and score.
>
> Best,
>
> AC

---

> ### Author Response · Authors · 2025-08-09
>
> Thanks so much, Reviewer WJP1 — we’re grateful for the opportunity to address your concerns during this rebuttal period, and we very much appreciate your feedback. We will certainly continue expanding SWE-smith's language coverage and utility, such that it hopefully serves as a useful repository of data for coding agent research.

---

### Official Review · Reviewer_puo9 · 2025-07-03

**Rating:** 6
**Confidence:** 5

**Summary:**

The paper presents **SWE-smith**, a scalable and efficient data collection pipeline for generating software engineering training datasets at the repository level. The authors introduce a novel approach that inverts the SWE-bench methodology: instead of finding issue instances first, SWE-smith creates execution environments for real-world Python repositories and synthesizes large numbers of realistic, test-breaking task instances through LM-generated and procedural code modifications. Using SWE-smith, the authors curate a dataset with over 50,000 instances from 128 repositories, representing a significant scale-up compared to prior work. The utility of this dataset is validated by training SWE-agent-LM-32B, which achieves state-of-the-art open-weight performance on the SWE-bench Verified benchmark (40.2% resolve rate). The entire pipeline, dataset, and models are open-sourced.

**Additional Feedback:**

### Questions for the Authors
1. Could you elaborate on how repository selection ensures diversity beyond popularity metrics (e.g., PyPI downloads, GitHub stars)? Are application domains (e.g., web dev, ML libraries, system tools) explicitly balanced?
2. You mention the collection strategy is transferable to other languages. Can you provide more concrete insights or preliminary results for languages like Java, C++, or JavaScript, where AST manipulation is less standardized?
3. Beyond fine-tuning, do you plan to investigate RL-based approaches, such as those explored in SWE-RL or FireAct, to better exploit SWE-smith for improving reasoning and exploration capabilities of software engineering agents?

**Dataset Code Accessibility:**

Yes

**Ethical Considerations:**

No, there are no or only very minor ethics concerns

**Limitations Weaknesses:**

1. **Language limitation**: The current pipeline is entirely Python-centric, relying on Python-specific AST tools. Although the authors claim transferability, no experiments or prototypes demonstrate generalization to other programming languages.
2. **Diversity of repositories**: While the dataset covers 128 repositories, the selection is filtered based on PyPI download counts and GitHub stars, which may bias the dataset toward popular, well-maintained libraries and underrepresent more complex or niche software domains.
3. **Agent-side limitations unaddressed**: The paper mainly evaluates the dataset through fine-tuning open LMs, but does not explore reinforcement learning or more advanced agent-side training techniques, which are crucial for maximizing the utility of such datasets in real-world agent systems.

**Strengths Contributions:**

1. **Scalable, practical contribution**: SWE-smith addresses a major bottleneck in software engineering agent development by significantly improving the scalability of high-quality, execution-validated task generation.
2. **Strong empirical results**: The approach leads to new open-weight state-of-the-art performance on SWE-bench Verified, with systematic ablation studies demonstrating the impact of different bug generation and issue synthesis strategies.
3. **Comprehensive and thoughtful design**: The combination of LM-generated bugs, procedural modifications, PR inversions, and automated issue generation reflects a well-engineered and labor-efficient pipeline that reduces human intervention compared to previous efforts like SWE-bench.

---

> ### Author Rebuttal · Authors · 2025-07-31
>
> # General Responses
>
> Our greatest thanks to the reviewers for their thoughtful, helpful feedback about SWE-smith; we very much appreciate the reviewers’ efforts and the strengths of our work they’ve identified.
>
> We identified 2 common concerns. To reduce redundancy, we’ve written this general response section to address these issues.
>
> ---
>
> ## 1) Bug Generation Techniques are limited to Python
>
> _Reviewer Feedback (PUO9; 4V7Q; WJP1): We do not show concrete evidence that our bug generation methods are transferable to non-python languages as claimed in Section 6._
>
> We’ve continued to expand language/repo support for SWE-smith in the open, and have made the following progress:
>
> * Added **100+ non-Python repos** (85 Golang, 11 JavaScript, 10 Rust, 1 for remaining 4 languages; 230+ total)
> * Added **9k Golang task instances** (59.1k total)
> * Most bug generation strategies (LM Generation, PR Mirrors, Combine Bugs) required no modifications and have been applied to **9 programming languages** so far (Python, Rust, C, C++, C#, Golang, Java, JavaScript, PHP)
> * The 4th bug generation strategy **(procedural modification) has been extended for Golang**.
>
> Insights about these efforts:
> * **Supporting new repositories is trivial**. Our process that uses SWE-agent to automatically construct a repo’s Docker image and define its specifications works out-of-the-box for non-Python repos.
> * **Supporting new languages is semi-trivial**. AST parsing support is available for several languages via the `tree-sitter-[lang]` PyPI packages. As a reminder, we use AST parsing to (a) identify programmatic entities (e.g., functions, classes), and (b) perform procedural modifications.
>     * For (a), `tree-sitter-[lang]` does much of the heavy lifting.
>     * For (b), some manual engineering + trial-and-error (~6 hrs for 10 Golang Procedural Mod. techniques) is required to ensure AST transformations can be recompiled back into well-formatted code.
> * **Creating non-Python task instances is trivial**. With the prior two augmentations, the existing bug and issue generation pipelines work as is.
>
> The SWE-smith GitHub repositories and HuggingFace assets have been updated to reflect these changes.
> We will add discussions of these initial efforts to the paper.
>
> With that said, creating full-fledged multilingual coding datasets and coding agents is, in our opinion, best left as follow up work.
> SWE-smith’s goal is to validate the concept of injecting bugs into a codebase as a viable task synthesis strategy.
> Leveraging this pipeline to understand how multiple programming languages affect training dynamics is fairly orthogonal, so it’s perhaps best left as a separate work.
>
> ---
>
> ## 2) Reinforcement Learning Experiments
>
> _Reviewer Feedback (PUO9; 4V7Q; DLBQ): In the paper, LMs are trained exclusively using supervised fine-tuning. We do not perform RL fine-tuning (e.g., PPO, GRPO) in any of our experiments._
>
> We agree with the reviewers that the RL training paradigm is very exciting, and showcasing SWE-smith’s viability for such settings would be quite timely.
> However, we’d like to highlight two points for why we did not include such experiments:
>
> _1) We want to focus SWE-smith’s narrative and contributions on the impacts of scaling data and data quality, not training algorithms._
> * In Section 4, we intentionally fix the training algorithm (rejection sampling fine-tuning / imitation learning) to be consistent with prior works (e.g., SWE-gym, R2E-gym), and only vary the data. Therefore, the **performance gains we report can be attributed directly to data quality and scale**.
>
> _2) Infrastructure for training LMs with online RL on multi-turn, agentic environments is an active research direction that is complementary but distinct. Such tools were not available as of SWE-smith’s release._
> * **Existing RL frameworks (e.g., TRL, VeRL) primarily target non-agentic tasks such as math reasoning**, which do not require extensive environment interactions. Adapting these frameworks to support complex interactions with SWE environments requires substantial engineering.
> * **Recent initiatives (e.g., [SkyRL](https://novasky-ai.notion.site/skyrl-v0), [Agentica](https://agentica-project.com/)) specifically aim to address these challenges**, underscoring that infrastructure for orchestrating interactions across multiple Docker containers in an RL training loop is an active, non-trivial, distinct research area.
> * Among prior work, the only demonstration of successful RL fine-tuning for SWE-bench is **SWE-RL**, which differs significantly in scope and setup:
>     * SWE-RL’s scaffold of choice **Agentless, which side-steps the need for execution or interaction**. SWE-smith’s goal is to unlock data for SWE-agent’s.
>     * SWE-RL’s data and training infra **have not been open-sourced** (the reward functions + Agentless mini scaffold are open). Reproduction is not possible without significant engineering effort to reproduce the paper.
>
> In summary, we acknowledge that **SWE-smith + RL would be quite worthwhile, but believe it'd be better suited as follow up work**.
> While we do not plan to incorporate these into the paper, we briefly highlight our relevant, ongoing efforts:
> * SWE-smith is actively being integrated into SkyRL, as reflected by the [Github repo](https://github.com/NovaSky-AI/SkyRL/branches)
> * We’ve released another 20k trajectories (on top of the original 5k trajs), which also includes trajectories corresponding to _unresolved_ task instances.
> * For this rebuttal, **we performed offline, DPO fine-tuning of SWE-agent-LM-7B on 500 pairs of trajectories**. Each pair of trajectories correspond to the same task instance; one trajectory successfully resolves the instance while the other does not. We observe a **+1.4% (13.1%)** improvement on SWE-bench Lite.
>
> ---
>
> ## 3) Summary of New Analyses
>
> We briefly review the new experiments and updates to our paper we plan to make in response to the reviewers’ helpful and constructive feedback.
>
> * **Appendix C**: Additional details about preliminary efforts to extend SWE-smith’s bug generation techniques beyond Python codebases.
> * [WIP] **Appendix D**: Additional qualitative comparisons of synthetic vs. real bugs (_Reviewer 4V7Q_)
> * **Section 4 / Appendix F**: Evaluation results on SWE-bench Live, Aider Polyglot, HumanEvalFix, SWE-bench Multilingual (_DLBQ_)
> * **Appendix F**: New ablation discussing effort of training on large vs. small GitHub repositories (_DLBQ_)
> * **Appendix F**: New ablation discussing effect of offline DPO fine-tuning on preference pairs of trajectories.
>
> ---
>
> # Response to Reviewer puo9
>
> We sincerely thank Reviewer PUO9 for highlighting SWE‑smith’s scalable pipeline, strong empirical gains, and thoughtful design, and for raising valuable questions about language generalization, repository diversity, and RL training.
>
> ---
>
> > W1: Language Limitation + Q2: You mention the collection strategy is transferable to other languages…
>
> **“General Response #1” should address these concerns.**
> We agree with your insight that AST manipulation is less standardized for some languages.
> * In practice, we found that `tree-sitter-[lang]` is quite reliable for identifying programmatic entities and constructing ASTs for many languages.
> * However, getting procedural modifications to work requires some additional engineering, as we experienced for Golang.
>
> ---
>
> > W2: Diversity of repositories + Q1: Could you elaborate on how repository selection ensures diversity beyond popularity metrics?
>
> Thank you to the reviewer for raising this point — it’s a valid concern, as we want SWE-smith to enable models that generalize meaningfully across diverse repositories.
>
> **We analyze domain diversity in Figure 3**, where we categorize the 128 repositories into six broad application areas. A complete list and brief descriptions are included in **Appendix D**. While we did not enforce category-level balancing during selection, the post-hoc categorization suggests that **filtering by stars and downloads captures a reasonably broad spectrum of real-world utilities**.
>
> We also found that popularity-based filters (stars/downloads) were effective for practical reasons: **repositories with higher usage tend to have more reliable environment specifications and tests**. In contrast, many low-star repositories lack usable tests, have brittle build processes, or are poorly maintained — making them less suitable for scalable data collection.
>
> Finally, **scale is an important factor that mitigates selection bias**. SWE-smith spans 128+ repositories — over 10x more than prior datasets — which offers much broader domain coverage than earlier efforts. No single repository represents more than 4% of SWE-smith task instances (`scanny__python-pptx.278b47b1` has the most, with 2,389 out of 59,131 total task instances).
>
> With that said, we think the reviewer makes a good point. **In SWE-smith’s recent expansion to Golang, we intentionally included repositories with fewer than 1,000 stars** (opposite of original condition), such as [`derekparker/trie`](https://github.com/derekparker/trie). SWE-smith was still able to generate valid bugs in such cases (e.g., [mirror repo](https://github.com/swesmith/derekparker__trie.4095f8e3)), though we note that low-star repositories often have smaller codebases, sparser test coverage, and limited documentation.
>
> ---
>
> > W3: Agent-side limitations unaddressed + Q3: Beyond fine-tuning, do you plan to investigate RL based approaches?
>
> **“General Response #2” should address these concerns.**
> Thanks for highlighting these works.
> * FireAct introduces the concept of fine-tuning on agent trajectories, which has become a widespread technique. We use FireAct and cite it in this work.
> * We discuss SWE-RL thoroughly in the general response.
>
> In summary, we agree that RL + SWE-smith is a promising and important direction, yet we believe it is better positioned as a distinct, follow-up work.

---

### Decision · Program_Chairs · 2025-09-18

**Decision:**

Accept (spotlight)

**Comment:**

The paper presents SWE-smith, a scalable and efficient data collection pipeline for generating software engineering training datasets at the repository level. The authors introduce a novel approach that inverts the SWE-bench methodology: instead of finding issue instances first, SWE-smith creates execution environments for real-world Python repositories and synthesizes large numbers of realistic, test-breaking task instances through LM-generated and procedural code modifications. Using SWE-smith, the authors curate a dataset with over 50,000 instances from 128 repositories, representing a significant scale-up compared to prior work. The utility of this dataset is validated by training SWE-agent-LM-32B, which achieves state-of-the-art open-weight performance on the SWE-bench Verified benchmark (40.2% resolve rate). The entire pipeline, dataset, and models are open-sourced.

After the rebuttal, the paper received 1 Strong Accept, 2 Accept and 1 Borderline Accept. The current limitations include the ongoing problem statement and the diversity of the programming languages. The authors are encouraged to keep improving the dataset on the mentioned aspects.

===== FINAL UPDATE FROM DB Track PCs ====

The final decision for this paper has been taken by the program chairs after consultation with the SACs. All Senior Area Chairs have ranked papers according to the feedback from the AC during the review process. We decided to leave the original meta-review to reflect the opinion of the AC in light of the initial discussions with reviewers and SAC.